# Combining cross-sectional and longitudinal genomic approaches to identify determinants of cognitive and physical decline

**Tabea Schoeler** [1,2,3] ✉, **Jean-Baptiste Pingault** [2,4] & **Zoltán Kutalik** [1,3,5] ✉

Large-scale genomic studies focusing on the genetic contribution to human aging have mostly relied on cross-sectional data. With the release of longitudinally curated aging phenotypes by the UK Biobank (UKBB), it is now possible to study aging over time at genome-wide scale. In this work, we evaluated the suitability of competing models of change in realistic simulation settings, performed genome-wide association scans on simulation-validated measures of age-related deweekcline, and followed up with LD-score regression and Mendelian Randomization (MR) analyses. Focusing on global cognitive and physical function, we observed marked differences between baseline function ($\theta$) and accelerated decline ($\Delta$). Both outcomes showed distinct heritability levels (e.g., 31.38% $h^2_\theta$ versus 3.15% $h^2_\Delta$ for physical function) and different associated loci (e.g., *DUSP6* specific to physical $\Delta$). Further, we found little commonalities across the two dimensions of aging—while cognitive decline was largely driven by Alzheimer's disease liability (standardized MR-effect, $\gamma = 0.17$), physical decline was mostly impacted by telomere length ($\gamma = -0.05$) and bone mineral density ($\gamma = -0.05$). Our work highlights the utility of longitudinal genomic efforts to scrutinize age-dependent genetic and environmental effects on physical and cognitive outcomes. Careful modelling and attention to participation characteristics are, however, crucial for valid inference.

Preserving cognitive and physical health in the global aging population is a major public health priority. Substantial resources are therefore invested into research scrutinizing modifiable risk factors of age-related decline[1,2], ultimately aiming to optimize preventative interventions that can slow down aging and delay the onset of functional impairment in the elderly. To that end, the genetic contribution to human aging is increasingly studied, holding promise to directly (e.g., via molecular/pharmacological targets) or indirectly (e.g., via environmental targets) provide novel insights into treatment and prevention. The emergence of large-scale biobanks in particular has moved the field forward, advancing our understanding of individual differences underlying indexes of aging (e.g., longevity[3], healthspan[4], frailty[5,6], biological aging[7–9]).

Although decline in cognitive and physical dimensions is considered part of normal aging, marked variation characterizes the rate and timing of age-related decline[10]. Yet, the genetic contribution to

[1]Department of Computational Biology, University of Lausanne, Lausanne, Switzerland. [2]Department of Clinical, Educational and Health Psychology, University College London, London, UK. [3]Swiss Institute of Bioinformatics, Lausanne, Switzerland. [4]Social, Genetic and Developmental Psychiatry Centre, Institute of Psychiatry, Psychology and Neuroscience, King's College London, London, UK. [5]University Center for Primary Care and Public Health, Lausanne, Switzerland. ✉e-mail: tabea.schoeler@unil.ch; zoltan.kutalik@unil.ch

such heterogeneity remains poorly understood, possibly echoing the limited power for genome-wide discovery in existing prospective cohorts[11–14], or potential biases affecting cross-sectional research on age-varying genetic effects (e.g., birth cohort effects[15] or age-dependent study participation[16]). Large-scale biobank initiatives committed to longitudinal assessments therefore represent an unprecedented resource for the study of aging processes, enabling a move from cross-sectional (e.g., level of grip strength) genome-wide analyses to an estimation of longitudinal (e.g., loss in grip strength from that point) genetic effects. For that purpose, the UK Biobank (UKBB) has curated and released a rich set of prospectively ascertained aging phenotypes[17], capturing key aspects of changes in physical (e.g., forced expiratory volume (FEV), fitness levels, grip strength) and cognitive (e.g., reaction time, fluid intelligence) dimensions. This resource now enables the investigation of prominent aging theories within a genetically informed framework, including common cause theories of aging[18–21] or the role cognitive/physical reserve in age-related decline[22–28] (c.f., 'Research in context' in the Supplement for further discussion).

Despite the promise as a powerful resource for the study of human aging, the analysis and interpretation of findings obtained from large prospective biobank samples present several challenges. First, selective participation already documented at the initial recruitment stage in the UKBB[29–31] may be exacerbated in longitudinal samples due to selective attrition and survival, where pro-spectively ascertained individuals represent an even healthier subset of the initially healthy[32]. Assessing the robustness of findings to selective attrition therefore constitutes a necessary step when making inferences from non-representative prospective samples. Second, while the recruitment of individuals with at least two-wave data (i.e., baseline and one follow-up) is moving towards its goal of 100,000 individuals in the UKBB[33], three-wave data (or more) is currently only available for a small fraction of the sample. As such, maximizing the number of individuals inevitably limits the number of follow-up waves, restricting longitudinal analyses to two-wave models of change that are not free of criticism[34]. While different change models have been proposed and applied in classic epidemiological[35,36] and genetic[37–40] studies, there is currently little consensus as to which approach (if any) is most appropriate for the discovery of longitudinal genetic effects. To minimize the risk of inferential errors, a deeper understanding of behaviours and limitations of commonly used two-wave models of change is therefore critical prior to their application in genome-wide scans.

In summary, this work aims to exploit the prospective UKBB resource to scrutinize genetic and environmental contributions to physical and cognitive aging. To that end, we propose and apply an analytical framework (c.f., Figs. 1 and 2 for illustration) tailored to the analysis of change at genome-wide scale, while accounting for design-challenges inherent to prospective volunteer samples.

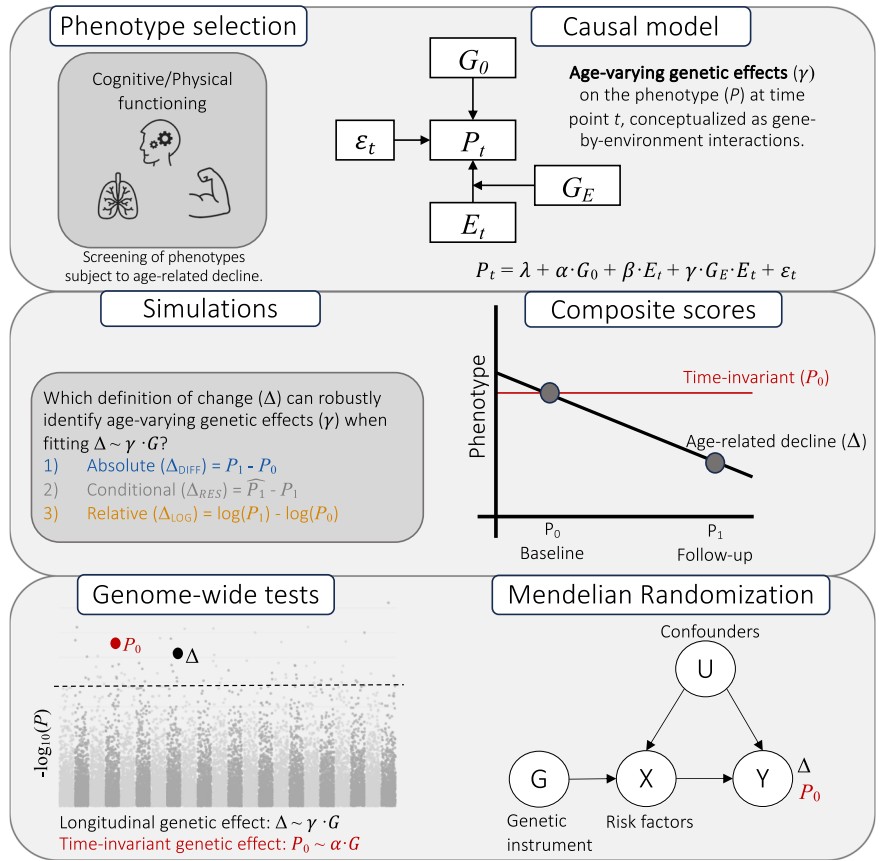

**Fig. 1 | Study framework.** Analytical framework to study longitudinal genetic effects on phenotypes subject to age-related decline. The structural causal model is shown at the top, where the phenotype $P$ at time point $t$ ($P_0$ = baseline, $P_1$ = follow-up) varies as a function of time-invariant (baseline) genetics ($G_0$), time-varying (longitudinal) genetics ($G_E$) and the environment ($E$). Simulations are used (middle panel) to assess the suitability of three definitions of change ($\Delta$) for longitudinal genome-wide analyses ($\Delta - \gamma \cdot G$), including absolute change ($\Delta_{DIFF}$, i.e., the absolute difference between the baseline phenotype, $P_0$, and the follow-up phenotype, $P_1$), relative change ($\Delta_{LOG}$, i.e., the difference between $\log(P_0)$ and $\log(P_1)$) and conditional change ($\Delta_{RES}$, i.e., the difference between the observed $P_1$ and the predicted $\hat{P}_1$ phenotype). Genome-wide tests and downstream analyses (bottom panel) are performed on composite scores of cross-sectional (i.e., time-invariant) and longitudinal (i.e., time-varying) indexes of cognitive and physical aging.

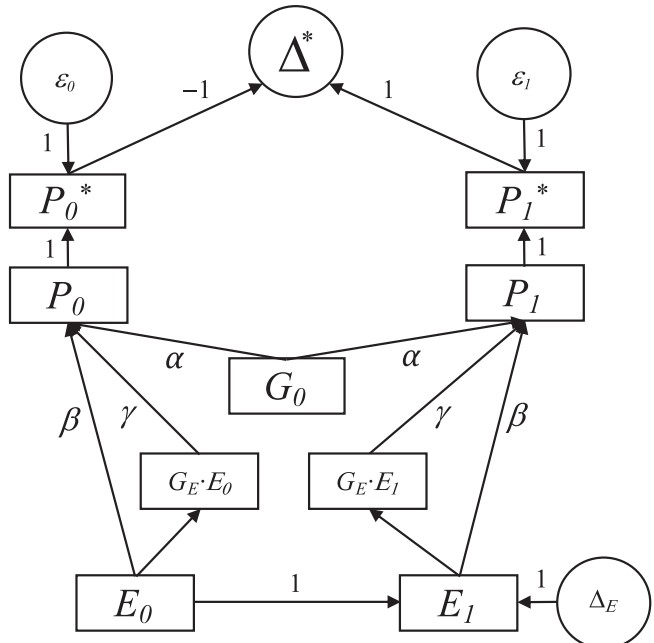

**Fig. 2 | Structural causal model of age-dependent genetic effects.** The observed phenotype $P^*$ at time point $t$ (0 = baseline, 1 = follow-up) was modelled as a function of time-invariant (baseline) genetics ($G_0$), time-varying (longitudinal) genetics ($G_E$) and the environment ($E$). All arrows labelled with Greek letters represent the structural coefficients of the underlying data-generating process, where $\alpha$ is the (main) baseline genetic effect, $\beta$ the (main) environmental effect and $\gamma$ the gene-by-environment effect. $\varepsilon_t$ is the measurement error in the phenotype.

## Results

### Indexes of age-related decline in cognitive and physical function

We first assessed if the selected measures of cognitive and physical function showed the expected age-related decline, by estimating the effects of age on each of the 17 aging phenotypes in univariate linear regression analyses. For that, the physical and cognitive measures were transformed into standardized z-scores ($\frac{P_0 - \mu_0}{\sigma_0}$, where $\mu_0$ and $\sigma_0$ are the mean and standard deviation of the baseline phenotype, $P_0$, respectively). Thereby, a negative $\alpha$-coefficient represents decline in standard deviations of a particular measure per additional year of age. As shown in Fig. 3, Supplementary Data 1 and Supplementary Fig. 1, all measures showed the expected age-related decline. However, the magnitude of age effects varied considerably across traits and UKBB sub-samples (mean $\alpha = -0.02$, ranging from −0.06 to −0.003). For example, steeper age-related decline was, on average, present for physical (mean $\alpha = -0.03$, $SE = 0.005$) than cognitive abilities (mean $\alpha = -0.02$, $SE = 0.005$). The largest effect of age was observed for psychomotor abilities (symbol digit substitution test), where the test performance decreased by -0.06 SD on average per additional year of age. Of note, while these results are interpreted as age effects, alternative factors (e.g., birth cohort effects[41]) may also contribute to the observed between-subject differences across age groups. Further, when examining UKBB sub-samples with varying degrees of representativeness, we found that more selective samples exhibited a less pronounced decline. As illustrated in Fig. 3, selective participation resulted in attenuated age effects (Panel A/B), where attenuation bias aggravated with increasing rates of loss to follow-up and non-representativeness (Panel C). The implications of selective participation and attrition are further discussed in the Supplement (Supplementary Discussion).

Slopes of global cognitive and physical decline were derived from 8 aging phenotypes, covering between 1 and 18 years of follow-up (mean = 7.8) (c.f., Supplementary Figs. 2–3 and Supplementary Data 2 for phenotype-specific characteristics). Complete data on global

cognitive and physical decline was available for 123,194 and 85,502 individuals, respectively. Supplementary Fig. 4 shows the correlations between the slopes of decline in cognitive and physical measures. While most slopes were positively correlated across the different aging phenotypes, the coefficients were substantially smaller compared to the correlations among the cross-sectional phenotype pairs (Supplementary Fig. 5).

Based on our simulation work (c.f., Supplementary Figs. 6–7 and Supplementary Discussion), we found that baseline-adjusted change ($\Delta_{RES}$) introduces bias by falsely associating cross-sectional (time-invariant) genetic effects with change. Conversely, models capturing absolute ($\Delta_{DIFF}$) and relative change ($\Delta_{LOG}$) are more robust in distinguishing cross-sectional from longitudinal genetic effects under various realistic conditions. In subsequent analyses, we therefore prioritized definitions of absolute and relative change when scrutinizing longitudinal genetic effects.

### Longitudinal and cross-sectional genetic variant effects on cognitive and physical function

Longitudinal ($\Delta_{LOG}$, $\Delta_{DIFF}$) and cross-sectional ($P_0$) genome-wide tests were performed on the composite scores indexing global physical and cognitive function, in addition to the 8 individual aging phenotypes used to derive the scores. In total, 10,338 independent genome-wide variants were identified, including 7 associated with longitudinal decline and 10,331 with cross-sectional function.

The top LD-independent variant associated with relative and absolute cognitive decline ($N = 103,938$) included a missense variant in *APOE* (rs429358), where each additional copy of the T allele showed protective effects on cognitive decline (e.g., $\beta_{LOG} = -0.03$, $P = 4.3\text{e-}19$). The same variant also reached genome-wide significance in tests on baseline function ($N = 404,449$), albeit smaller effect magnitude (increasing cognitive function, with $\beta_{P_0} = 0.02$, $P = 3.2\text{e-}11$) (Fig. 4). rs117041440 (closest gene: *KLF4*) showed more $\Delta$-specific effects, associating with reduced decline in fluid intelligence ($\beta_{LOG} = -0.07$, $P = 1.9\text{e-}08$) but not baseline fluid intelligence at genome-wide significance ($\beta_{P0} = -0.02$, $P = 0.036$) (Supplementary Fig. 8).

One variant (rs113645269, closest gene: *DUSP6*) was identified for global (relative) physical decline, increasing physical decline by $\beta_{LOG} = 0.09$ standard deviations per additional copy of the G allele ($P = 2.5\text{e-}08$, $N = 72,220$). This variant was specific to decline and did not impact baseline physical function ($P = 0.6$, $N = 405,979$). Similarly, rs190141474 (closest gene: *MNX1*) associated with less relative decline in FEV ($\beta_{LOG} = -0.14$, $P = 2.3\text{e-}09$, $N = 44,048$) but not baseline FEV ($P = 0.32$, $N = 373,397$). Both variants have no documented genome-wide associations with previously studied traits (c.f., PheWAS plots, Supplementary Fig. 9).

Among the six phenotypes assessed in interaction testing (cross-sectional physical and cognitive function, absolute cognitive and physical decline, and relative cognitive and physical decline), two genetic variants exhibited sex-specific effects: rs13141641 (nearest gene: *HHIP*) and rs9748016 (*RFLNB*), both of which were identified in the genome-wide tests on baseline physical function. Mapping these variants to prior phenotype-genotype associations revealed that these sex-dependent genetic effects mostly linked to indices of physical health, such as lung function and heel bone mineral density (Supplementary Fig. 10).

We found only negligible $h^2$-estimates on measures of decline, ranging from 0.03% to 1.2% for measures of cognitive decline and 0.98% to 3.15% for measures of physical decline. Significant $h^2$-estimates ($P < 0.05$) on change were present for global physical decline ($h^2_{DIFF} = 3.2\%$ and $h^2_{LOG} = 2.7\%$), decline in height ($h^2_{DIFF} = 2.4\%$ and $h^2_{LOG} = 2.3\%$), decline in grip strength ($h^2_{DIFF} = 1.4\%$), decline in FEV ($h^2_{DIFF} = 2.2\%$) and decline in fluid intelligence ($h^2_{DIFF} = 0.9\%$). Heritability estimates obtained for measures of baseline functioning were

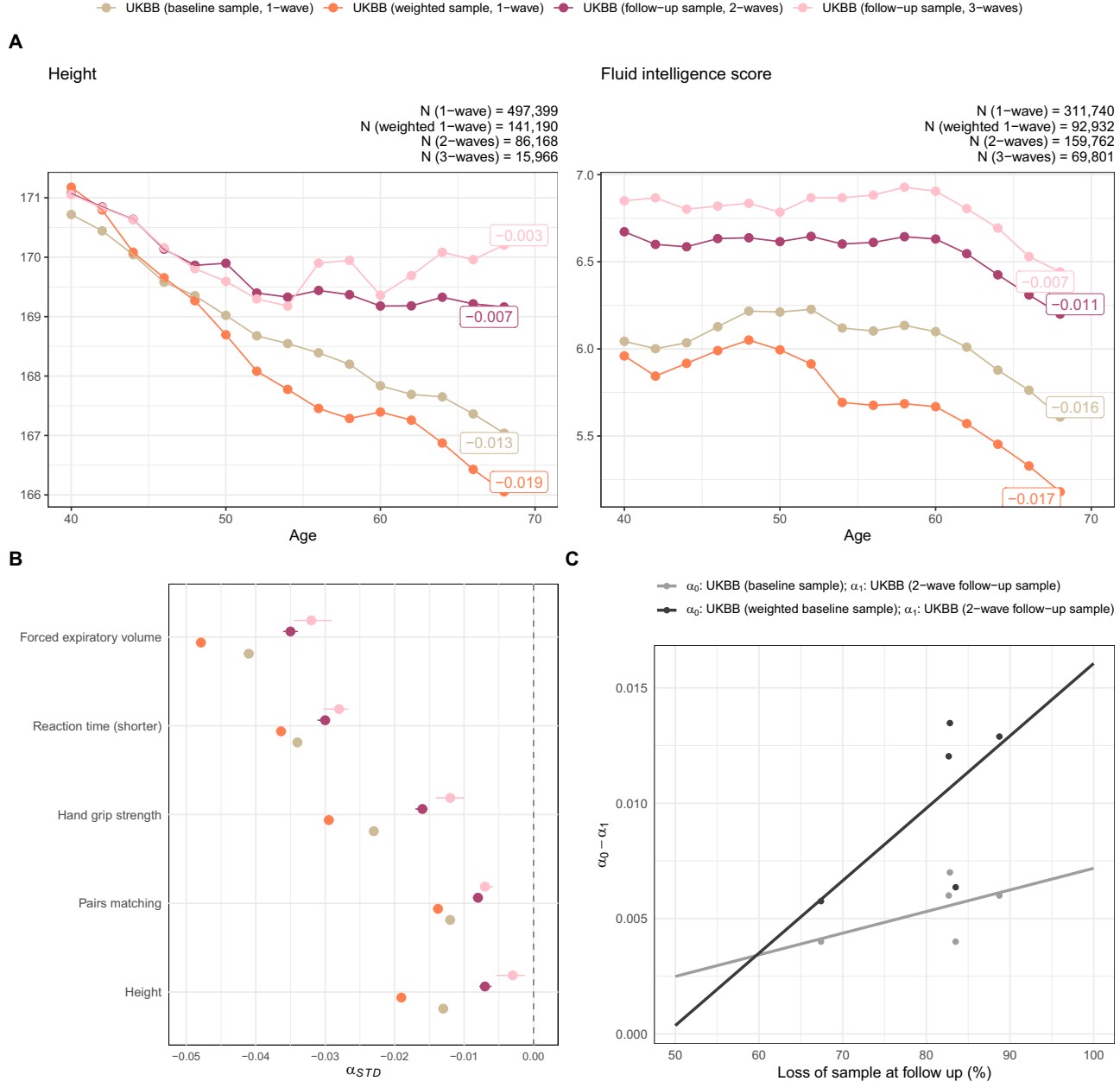

**Fig. 3 | Age-related decline in measures of cognitive and physical function.**
**A** Each dot represents the mean phenotype score assessed at baseline (*y*-axis) per 2 year age bin (*x*-axis) across UKBB samples with varying levels of representativeness. Age effects (c.f., labels to the right) on cognitive and physical measures assessed at baseline were obtained from linear regression models, applied in (a) the unweighted UKBB baseline sample ('baseline sample, 1-wave'), (b) the inverse probability weighted UKBB baseline sample ('weighted 1-wave'), (c) the UKBB follow-up sample with complete data in at least one follow-up assessment ('follow-up sample, 2-waves') and (d) the UKBB follow-up sample with complete data in at least two follow-up assessments ('follow-up sample, 3-waves'). Individual plots for all cognitive and physical decline phenotypes are included in Supplement

Supplement Tables 2--3. **B** Summary of age effects (with 95% confidence intervals) on cognitive and physical baseline phenotypes, obtained from the four UKBB sub-samples (a-d). Phenotypes shown in this figure include those with little missing data when assessed at baseline (i.e., at least 450,000 UKBB participants). The negative $\alpha_{STD}$-coefficient represents decline in standard deviations in the outcome per additional year of age. **C** plots the differences in age effects (y-axis) obtained from the baseline UKBB sample ($\alpha_0$, unweighted baseline sample in grey, weighted baseline sample in black) and the UKBB follow-up sample ($\alpha_1$) for the phenotypes shown in (**B**). The x-axis indicates the attrition rate per phenotype. The two plotted lines are the lines of best fit. Source data are provided in Supplementary Data 1.

larger in all instances (Fig. 4), ranging from 13.5% – 52.7% for physical function, and from 6.6% – 20.9% for cognitive function.

In line with our simulation results, longitudinal genetic estimates obtained from baseline-adjusted change ($\Delta_{RES}$) falsely captured substantial parts of the baseline genetic effects, leading to an inflation in genomic signal (Supplementary Fig. 8, in grey). Here, the impact of adjustment-bias was particularly prominent for global cognitive

decline, evident by the excess in genome-wide identified variants ($k_{RES} = 6$ versus $k_{LOG} = 1$) and SNP-heritability ($h^2_{RES} = 6.3\%$ versus $h^2_{LOG} = 0.2\%$). A complete summary of the genome-wide results is included in Supplementart Data 3.

Finally, we performed weighted genome-wide association analyses to correct for possible bias resulting from selective follow-up participation. Supplementary Fig. 11 shows that bias was present in

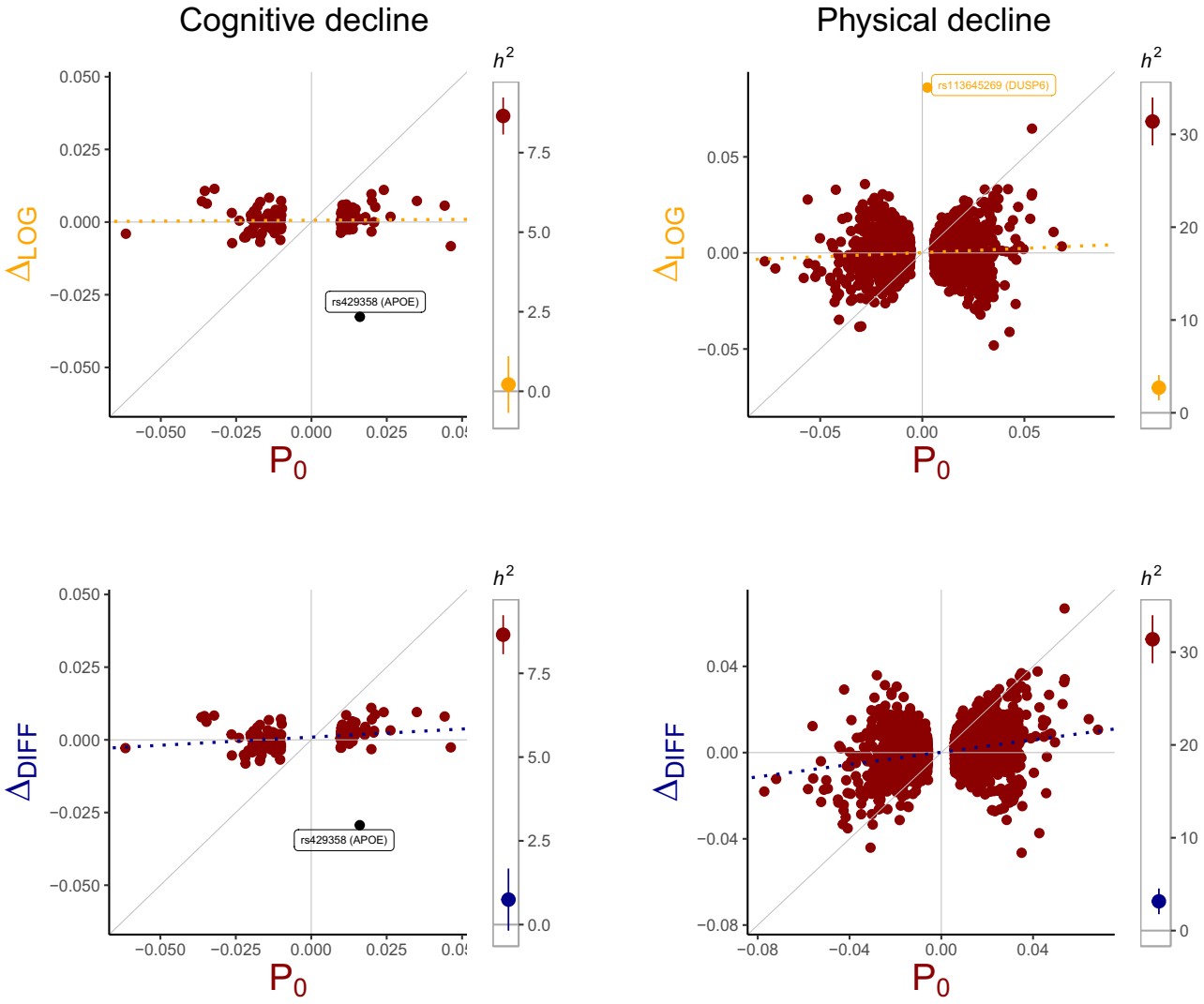

**Fig. 4 | Cross-sectional and longitudinal genetic effects on physical and cognitive function.** Effects estimates of variants reaching genome-wide significance in association tests on either cross-sectional physical/cognitive functioning (x-axis) or cognitive/physical decline (y-axis). The colour scheme highlights variants associated with either cross-sectional ($P_0$ in red), longitudinal ($\Delta_{LOG}$ in orange, $\Delta_{DIFF}$ in blue) or both (in black) outcomes. The dashed slope (line of best fit, obtained from $\beta_\Delta$ - $\beta_{P0}$) represents the association between the cross-sectional and longitudinal SNP effects. The SNP-heritability estimates ($h^2$, with 95% confidence intervals) obtained from longitudinal and cross-sectional genome-wide analyses are shown to the right of each panel. Source data are provided in Supplementary Data 3.

both directions, leading to over-estimation (e.g., *APOE*-effects on cognitive-Δ) and under-estimation (e.g., *DUSP6*-effects on physical-Δ) of variant effects. Across all tested variants, we did not, however, find evidence of altered direction of effects.

### Risk factors of longitudinal and cross-sectional cognitive and physical function

Mendelian Randomization (MR) analysis was used to identify causal factors involved in age-related decline. Contrasting results from phenotypic association tests to MR-estimates (c.f., Supplementary Fig. 12) indicated that many factors reaching significance in phenotypic analyses (133 out of 228 exposure-outcome associations with $P < 0.05$) showed smaller and non-significant effects when tested in MR, possibly reflecting the influence of confounders and reduced statistical power. More specifically, only few exposures showed substantial causal effects on decline (e.g., shorter parental lifespan), and more common were influences of smaller magnitude that were specific to either cognitive or physical decline (Fig. 5). For example, while cognitive decline was mostly predicted (at nominal significance level) by

Alzheimer's disease, lipid traits (e.g., Apolipoprotein A and B) and behaviours potentially altering those (e.g., vegetable intake), there was little overlap with risks identified for physical decline (e.g., shorter telomere length, higher bone mineral density, basal metabolic rate, poor sleep). 10 exposures on cognitive/physical decline survived suggestive Bonferroni correction for multiple testing ($P < 0.05/11 = 0.005$, were 11 is the number of independent exposure dimensions) (c.f., triangle shapes in Fig. 5), of which 6 (highlighted in rectangular shapes) remained significant following stringent Bonferroni correction ($P < 0.05/106 = 0.0005$, were 106 is the number of exposures tested in MR). Of note, most of the stringently identified risk factors pointed towards scale-dependent conclusions. For example, higher baseline physical function predicted steeper physical decline when assessed in absolute ($\gamma = 0.1$, $P = 8.1e-14$) but not relative ($\gamma = 0.02$, $P = 0.16$) terms, a pattern consistent with the presence of non-linear change. Further, parental lifespan (PLS) on cognitive decline was flagged as possibly biased based on a number of MR sensitivity analyses, given the level of heterogeneity (Q-test $P = 3.2e-08$), the significant MR-Egger intercept term ($P = 1.5e-05$ for $PLS \rightarrow \Delta_{LOG}$) and the non-significant MR-PRESSO

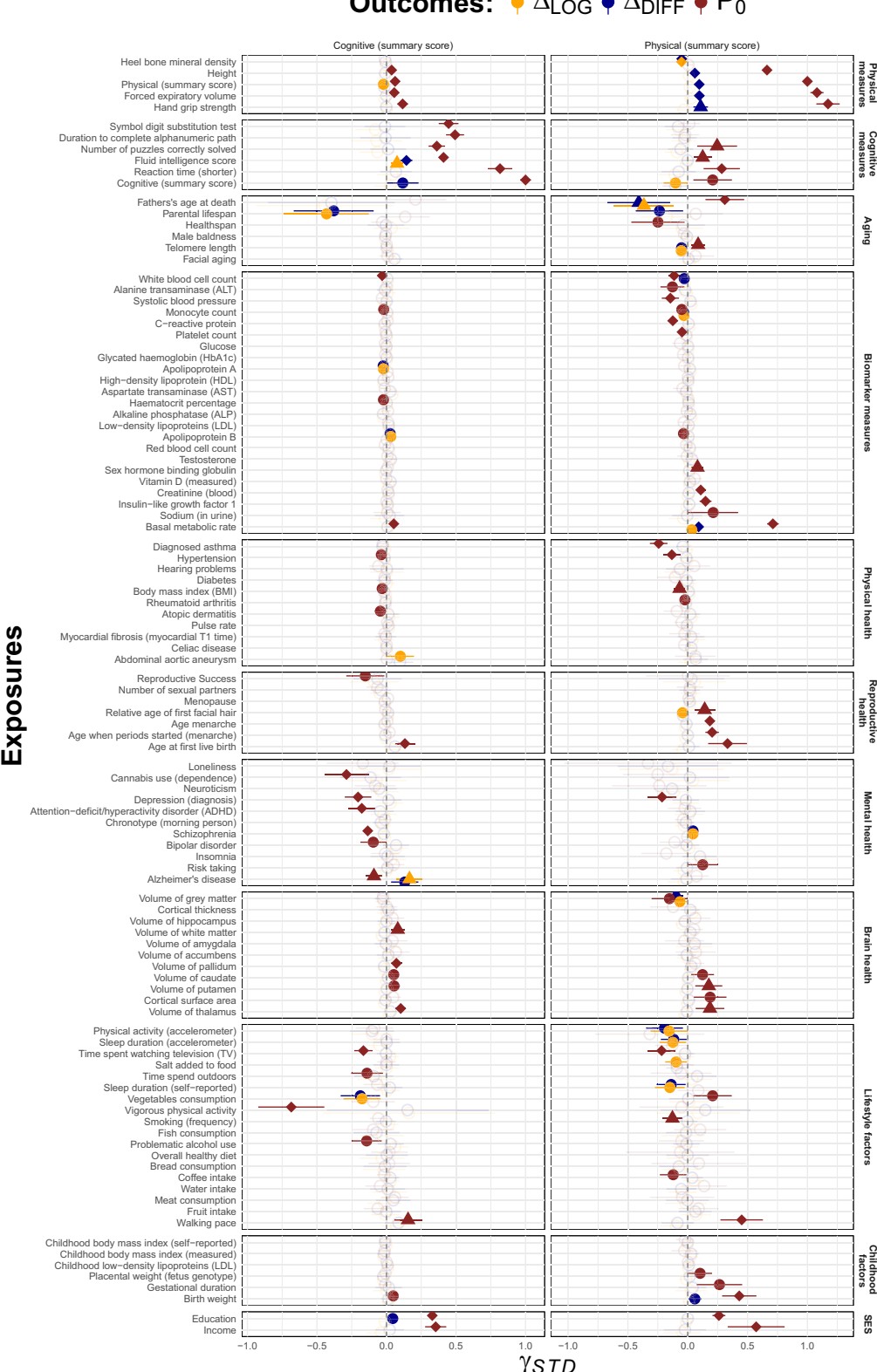

**Fig. 5 | Risk factors of longitudinal and cross-sectional cognitive and physical aging.** Standardized Mendelian Randomization effect estimates ($\beta_{STD}$, with 95% confidence intervals) of exposure effects on cross-sectional outcomes ($P_0$ in red, with positive coefficients indexing higher levels of function) and longitudinal outcomes ($\Delta_{DIFF}$ in blue and $\Delta_{LOG}$ in orange, with positive coefficients indexing larger decline). Filled points, triangles and diamonds highlight significant effects (at $P < 0.05$, $P < 0.05/11$ and $P < 0.05/106$, respectively). Circles highlight non-significant ($P > 0.05$) effects. Source data are provided in Supplementary Data 4.

effects (e.g., $P = 0.23$). The complete set of MR-results, including the exposure-outcome associations when performing MR on the individual aging phenotypes, is included in Supplementary Data 4 and Supplementary Figs. 13–14.

Further, we found little evidence of common underpinnings shared by (cross-sectional) level of functioning and (longitudinal) decline in cognitive and physical outcomes, implicated by the lack of association between cross-sectional MR-effects ($\gamma_{P0}$) and longitudinal MR-effects when modelled as relative change [Pearson correlation coefficient $r(\gamma_{P0}, \gamma_{LOG}) = -0.06$, $P = 0.56$] or absolute change [$r(\gamma_{P0}, \gamma_{DIFF}) = 0.03$, $P = 0.72$]. Shared liabilities were, however, observed across cognitive and physical domains within cross-sectional analyses, notably for risks relating to lifestyles (e.g., time spent watching television), brain health (e.g., volume of thalamus) and social factors (e.g., income, education). Other exposures appeared more specific with respect to the outcome dimension, where risk factors tapping into mental health (e.g., ADHD, bipolar disorder, schizophrenia) were more specific to cognitive function, and biomarker traits (e.g., creatinine levels, sex hormone binding globulin, insulin-like growth factor 1) mostly to specific physical function.

## Discussion

Large-scale longitudinal biobank samples such as the UK Biobank (UKBB) have the potential to advance our understanding of the genetic and environmental contributions to aging. In this work, we exploited the prospectively ascertained UKBB sample to separate cross-sectional from longitudinal cognitive and physical function within a genome-wide framework, while evaluating the impact of design-constraints inherent to longitudinal biobank schemes. A more comprehensive discussion complementing the findings described below is available in the section 'Research in Context' in the Supplement.

We first evaluated three commonly used definitions of change for two-wave prospective data (i.e., absolute, conditional, relative change). Simulation work implicated that relative and absolute change can robustly distinguish longitudinal from cross-sectional genetic effects in realistic simulation settings, without showing susceptibility to biases affecting baseline-adjusted change scores. Exacerbation of false-positives via baseline adjustment occurs in situations where a genetic variant links to the baseline phenotype, resulting in bias that is proportional to the baseline genetic effects and the measurement error in the phenotype. Our recommendations are therefore in line with existing discussions[37–39] in discouraging the use of baseline-adjusted change for longitudinal genetic effect estimation. With respect to absolute/relative change, inconsistent results (e.g., male sex linking to increased absolute but decreased relative physical decline) can occur in situations where change is non-linear (e.g., exponential). The choice between modelling absolute vs relative change will therefore alter the interpretation of the results: If the priority lies in isolating change from possible baseline dependencies, approaches of relative change can help to account for non-linear change. Alternatively, absolute change may be preferred if a quantification of change on the original scale is of particular relevance (e.g., variant effects on kg-change following intervention[42,43]). More comprehensively, researchers may choose to model both absolute and relative change, to facilitate the interpretation of identified variant effects and identify possible scale-dependencies.

Moving forward with models of relative/absolute change, we performed genome-wide tests to assess the genetic contribution to cross-sectional and longitudinal phenotypes of aging. *APOE* was identified as the top gene associated with accelerated cognitive decline and, to a lesser extend, with lower levels of cognitive function. Since *APOE* has already been linked to numerous measures of age-related change and cross-sectional functioning (e.g., cognitive ability[44–47]/decline[14,44,45,47,48], brain levels/change[49], Alzheimer's disease risk/progression[50,51], BMI levels/change[52,53], biomarker levels/

change[38,54]), together these findings highlight that *APOE* is neither trait nor state specific. Together with evidence from MR implicating lipid traits as causal factors involved in cognitive decline (further discussed below), focusing further on lipids may therefore prove a promising avenue for future research into cognitive health[55–57]. *DUSP6* (Dual Specificity Phosphatase 6) and *MNX1* (Motor neuron and pancreas homeobox 1) were implicated in different proxies of physical decline. Both genes are commonly studied in the context of cancer progression (e.g., pancreatic[58], lung[59], colorectal[60], leukemia[61]) and immune system responses[62], suggesting that these variants may increase sensitivity to environmental stressors. More broadly, the identified variant effects may represent gene-environment interactions, where the genetic effects differ across changing environments as individuals age. While gene-by-environment studies typically examine interactions of a genotype with defined environmental exposures (e.g., pharmacological intervention[38], lifestyles[63–65], environmental toxicants[64]), here we capture (any) environmental change occurring during the aging process. Leveraging prospective data to examine gene-by-age effects in this context offers a particularly robust solution, which avoids common sources of bias (e.g., birth cohort effects) that complicate the interpretation of cross-sectional interaction effects.

Despite the relatively large sample size, only few genetic variants showed significant age-dependent effects. While this could reflect insufficient power to detect small effects, our findings point towards a lack of major genetic influences on within-individual decline. For example, cross-sectional cognitive and physical function showed orders of magnitude higher levels of SNP-heritability compared to age-related decline, consistent with what would be expected based on previous genome-wide works on within-individual change (e.g., $h^2 < 5\%$ for BMI[52,53], brain function[49], drug response[38]). As such, our results suggest that unmodelled gene-by-age interactions are unlikely to account for substantial portions of missing trait-heritability and population heterogeneity among cognitive and physical measures of aging. Of note, while indexes of decline likely contain more measurement error than the baseline measures from which they are derived[66], thereby inducing downward bias in $h^2$-estimates[67], this attenuation bias is unlikely to fully explain the stark differences in variance components (e.g., $h^2$ for baseline height = 52.71% versus $h^2_\Delta$ for reductions in height = 2.35%). Instead, our observations imply that environmental factors may play a more significant role in driving within-person aging processes, in line with twin-study evidence attributing mostly non-shared environmental sources to variations in the rate of change[68,69].

Mendelian Randomization was used to further disentangle risk factors of accelerated decline, defined as a more rapid deterioration in physical and cognitive function over time, yielding three main insights: First, we observed little commonalities across the two dimensions of decline - while Alzheimer's disease liability and shorter parental lifespan constituted the main risk factors of accelerated cognitive decline, physical decline was predicted by a number of biological factors amenable to lifestyle and/or environmental interventions (e.g., basal metabolic rate, telomere length, bone mineral density). This contrasts findings from cross-sectional MR, where levels of cognitive and physical function showed shared aetiologies with overall larger exposure effects. Second, our findings implicate that high baseline function in a given trait (e.g., cognition), or factors shaping that trait (e.g., educational attainment), may not serve as a buffer against age-related decline in that trait. In other words, while high cognitive or physical reserve can delay the onset of functional impairment, it is unlikely to provide additional protective effects by slowing the rate of decline, in line with previous observations[70–76].

The presented results should be interpreted in light of a number of limitations (see also Supplement for an extended discussion on the study limitations). First, age-related change was assessed using data from only two time points per individual, which reduces measurement precision compared to approaches utilizing more intensive

longitudinal data. Second, despite the large sample size, statistical power likely remains an issue. Genetic interaction effects are inherently small and harder to detect compared to marginal effects[77], which may have hindered the identification of genetic variants and causal factors of decline. Third, the predictability of age-related decline was generally low, given the number and effect sizes of the exposures identified in MR. Since our genetically informed framework only tests for lifetime risk factors, non-instrumentable time-varying environmental exposures previously implicated in age-related decline are therefore not captured in this work, examples of which include life events (e.g., loss of spouse[78]), age-related biological changes (menopausal status[79]), toxic environmental exposures (e.g., air pollution[80]) or changes in medication use[81]. As such, while barriers to risk-identification may reflect insufficient power to detect small exposure effects in MR, an alternative scenario is that unmodelled time-varying environmental factors play an important role in aging processes. Finally, implementing strategies designed to increase sample representativeness (Inverse Probability Weighting, IPW), we explored the impact and direction of bias resulting from selective participation. The results implicated that selective participation influenced both phenotypic and genotypic estimates (c.f., Supplementary Discussion). Therefore, boosting retention rates in future follow-up assessments will be crucial to minimize existing attrition biases. This is particularly important as statistical tools designed to probe the robustness of findings (e.g., IPW) reach a bottleneck in highly non-representative sample, where genome-wide discovery is hampered by the substantial loss in (effective) sample size when performing bias-corrected genome-wide tests.

In summary, our findings suggest that largely distinct genetic and environmental mechanisms characterize levels of function and decline in indexes of physical and cognitive function. Combining cross-sectional with longitudinal genomic efforts therefore holds promise for discovery of additional preventative targets that can help push back the age at which functional impairment begins. Importantly, design and analytical limitations of longitudinal genomic efforts can threaten the validity of findings, such as inferential errors resulting from inappropriate modelling of change or bias induced by selective attrition. While these constraints should ideally be addressed at the design stage of a study, here we provide a conceptual and analytical framework to guide longitudinal genomic initiatives when these conditions are not met. We conclude that large-scale prospective biobank data represent a powerful resource that comes with unique opportunities and challenges when studying the genetic basis of dynamic processes.

## Methods

### UK Biobank longitudinal assessments and slopes of age-related decline

The methodological framework of this work is illustrated in Fig. 1. We used data from the UK Biobank (UKBB), a large population-based cohort of >500,000 adults aged between 40–69 at baseline. The UKBB was approved by the National Health Service North-West Center Research Ethics Committee (REC No. 16/NW/0274) and this research was conducted under application number 16389. Between 2006 and 2010, UKBB participants completed a battery of cognitive and physical tests when attending the baseline assessment in one of the 22 research centres located across the United Kingdom. Since then, most tests were re-administered at varying follow-up intervals and in different subsets of the UKBB. The first repeat assessment centre visit took place (on average) in 2013 and included around 20,000 participants living within 35 km of the Stockport Biobank coordinating centre (21% response rate to the email/letter invitation[82]). The second repeat assessment centre visit (ongoing) started in 2014 and includes the first brain magnetic resonance imaging assessment, inviting back around 100,000 of the original volunteers[83]. The most recent assessment centre visit

(ongoing) aims to re-invite 60,000 individuals with existing imaging data to take part in repeat imaging between 2022 and 2028[84].

Supplementary Fig. 15 lists the 17 aging phenotypes initially selected for longitudinal analyses in this work, plotting the number of available follow-up assessments per measure, the environment in which it was obtained (assessment centre versus online) and the number of participants with complete multi-wave data. Of note, despite the availability of a number of self-report measures relevant for aging (e.g., walking pace, physical activity), we only included objectively ascertained phenotypes. As self-report measures are prone to (age-related) misreporting and measurement error[67,85,86], which reduces power for gene discovery and creates challenges for models of change[87], these measures were not included in this work. In total, we selected 6 physical measures and 11 cognitive measures (c.f., Supplement for a more detailed description). Supplementary Fig. 15 and Supplementary Data 5 list the number of participants with non-missing longitudinal data per phenotype. While up to four waves of data (i.e., baseline and three follow-up assessments) are collected, complete four-wave data was only available for a small fraction of individuals (between 19 to 18,917 participants, mean = 2654) and was therefore not considered for longitudinal genome-wide analyses. Similarly, since the number of individuals with complete three-wave data was small (between 527 and 69,801 participants, mean = 16,350), we restricted all longitudinal analyses to individuals with non-missing two-wave data (up to 161,821 individuals, mean = 66,655). Of note, as three-wave data for fluid intelligence was available for a relatively large subset (69,801 participants), we explored possible gains resulting from adding one more time point (c.f., Supplementary Discussion).

Moving forward with two-wave models of change to derive the individual indexes of decline, we evaluated (c.f., simulation work in the Supplement), applied and compared the results obtained from three competing definitions of change[38], including (1) difference scores ($\Delta_{DIFF}$), (2) residual change scores ($\Delta_{RES}$) and (3) log-difference scores ($\Delta_{LOG}$). All change scores were adjusted for the follow-up duration (details below) and computed such that higher values represent increased age-related decline:

(A) Absolute change, using difference scores ($\Delta_{DIFF}$), derived by subtracting the follow-up phenotype from the baseline phenotype ($\Delta_{DIFF} = P_0 - P_1$)

(B) Conditional change, using residual change scores ($\Delta_{RES}$), derived by regressing the baseline phenotype out of $\Delta_{DIFF}$, thereby indexing absolute change that is no longer predictable by the observed baseline scores.

(C) Relative change, using log-difference scores ($\Delta_{LOG}$), computed as the difference between the natural log-transformed baseline and follow-up phenotype ($\Delta_{LOG} = log(P_0) - log(P_1)$).

Indexes of decline ($\Delta$) were generated for all aging phenotypes meeting the criteria for inclusion: First, we included only phenotypes with at least 40,000 non-missing longitudinal (i.e., two-wave) observations. Second, to avoid problems resulting from poor measurement reliability, for example as documented for some of the cognitive measures[88] in the UKBB, we also discarded phenotypes showing poor consistency across time (i.e., $r < 0.4$, where $r$ is the average correlation across measurement occasions per phenotype, c.f., Supplementary Figs. 16–17). The box and and scatter plots of the selected cognitive and physical measures are shown in Supplementary Figs. 2–3. For all selected phenotypes, we then extracted the residuals from a model regressing change ($\Delta$, defined as either $\Delta_{DIFF}$, $\Delta_{RES}$ or $\Delta_{LOG}$) on age and the follow-up (FU) duration ($\Delta = age_0 + FU + age_0 \times FU + age_0^2$). FU was defined as $FU = age_1 - age_0$, where $age_0$ and $age_1$ correspond to the age at which $P_0$ and $P_1$ were obtained, respectively. $age_0^2$ was included in our models to account for possible non-linear rates of change, i.e., situations where the effect of age on within-individual change increases over time. In the last step, the composite scores for global physical

and cognitive decline were computed by averaging the standardized Δ-scores (mean = 0 and s.d. = 1), where we used row-wise mean imputation to address missing values.

To compare longitudinal to cross-sectional findings, we also derived composite scores of global cognitive and physical baseline functioning, using the same protocol as described above. In brief, we included the same set of aging phenotypes, extracted the baseline data of each phenotype and averaged the standardized baseline-scores (residualized for age and age$^2$). The resulting composite scores for global physical and cognitive function ($P_0$) and decline (Δ) were used as the primary outcomes in subsequent phenotypic, genome-wide and downstream analyses.

We performed a series of simulation analyses to assess risk of bias when using the three change definitions described above, including absolute ($\Delta_{DIFF}$), conditional ($\Delta_{RES}$) and relative change ($\Delta_{LOG}$). We simulated data according to the structural causal model shown in Fig. 2. As illustrated, the longitudinal (i.e., age-dependent) effects were conceptualized as gene × environment interactions, where the genetic effect differs across age-varying environments. The observed phenotype $P^*$ at time point $t$ (0 = baseline, 1 = follow-up) was modelled as follows:

$$P_t^* = P_t + \varepsilon_t = \lambda + \alpha \cdot G_0 + \beta \cdot E_t + \gamma \cdot G_E \cdot E_t + \varepsilon_t \quad (1)$$

where $P_t$ is the phenotype free of measurement error, $\varepsilon_t$ the measurement error in the phenotype, $\lambda$ the intercept, $\alpha$ is the time-invariant (cross-sectional) genetic effect, $G_0$ the time-invariant (cross-sectional) genetics, $\beta$ the environmental effect, $E$ the environment, $\gamma$ the gene × environment effect, $G_E$ the time-varying (longitudinal) genetics. A detailed description of the simulation approach is included in the Supplement.

**Statistics & Reproducibility.** No statistical method was used to predetermine sample size. Details regarding the sample selection process for the cross-sectional and longitudinal analyses are provided in Supplementary Fig. 18. This analyses are reproducible from the publicly available analytical scripts.

## Longitudinal and cross-sectional genome-wide scans and downstream analyses

We then performed longitudinal genome-wide association (GWA) tests on indexes of cognitive and physical decline and compared the results to GWA findings obtained on baseline (cross-sectional) physical and cognitive function. Tests were performed on the cognitive and physical decline composite scores, as well as the individual indicators used to derive the composite scores. In total, we tested four models per aging phenotype:

(1) Longitudinal genetic effects (difference scores): $\Delta_{DIFF} = \beta \cdot G + \varepsilon$
(2) Longitudinal genetic effects (residual change scores): $\Delta_{RES} = \beta \cdot G + \varepsilon$
(3) Longitudinal genetic effects (log-difference scores): $\Delta_{LOG} = \beta \cdot G + \varepsilon$
(4) Cross-sectional genetic effect (baseline phenotype, $P_0$): $P_0 = \beta \cdot G + \varepsilon$.

All models were adjusted for the first 10 genetic principal components (PCs), genotyping array, and sex. Age was not included as a covariate as all outcomes were residualized for age and age$^2$ prior to inclusion in genome-wide analyses. Genome-wide scans were performed in REGENIE (v3.2.6)[89], which proceeds in two main steps: Step 1 fits a whole-genome regression model via ridge regression to account for local LD structures and relatedness. QC-filtered SNPs were included in step 1 (minor allele frequency (MAF) ≥ 1%, Hardy-Weinberg equilibrium $P$-value ≥ 1 × 0$^{-15}$, genotyping rate ≥ 99%, not involved in inter-chromosomal LD and passing LD pruning at $R^2$-threshold of 0.9 with a window size of 1000 markers and a step size of 100 markers[89]). In step

2, the association test is carried out by fitting linear regression models conditioning on the predictions derived in the first step. This step uses the imputed UKBB genotypes (version 3, imputed using the Haplotype Reference Consortium) with MAF > 1%. The sample was restricted to individuals of European genetic ancestry and individuals with high missing rate and/or high heterozygosity on autosomes (as determined by the UKBB[90]) were excluded. As age-related change in cognitive and physical function have been reported to differ between sexes[91,92], we also estimated sex-dependent genome-wide effects. For that, we extended our models on change and baseline level of function by including a gene-by-sex interaction term (Δ = $\beta_1 \cdot G + \beta_2 \cdot G \cdot SEX + \varepsilon$ and $P_0 = \beta_1 \cdot G + \beta_2 \cdot G \cdot SEX + \varepsilon$, respectively).

LD-independent SNPs reaching genome-wide significance ($P < 5 \times 10^{-8}$) were selected via clumping in plink[93] (−clump-kb 250 −clump-r2 0.1). Lead SNPs were annotated to the nearest gene and mapped to previously associated phenotypes using the Open Target Genetics R package *otargen*[94]. The same software was used to map lead SNPs to previously studied phenotypes, by drawing statistical summary data from public databases such as UKBB (www.nealelab.is), FINNGEN and GWAS Catalogue (https://www.ebi.ac.uk/gwas/). SNP-heritabiliy ($h^2$) estimates were obtained using LD-score regression[95] as implemented in *GenomicSEM*[96].

We employed Mendelian Randomization (MR) analysis to evaluate causality of selected exposures with cross-sectional and longitudinal cognitive/physical outcomes. MR uses genetic instrumental variables (IV) as proxies for the exposure of interest (e.g., genetically predicted exposure to smoking), to estimate the causal effect of the exposure on an outcome[97]. We included data from 144 genome-wide studies to extract the genetic instruments, tapping into 11 exposure dimensions, such as aging-markers (e.g., telomere length), biomarkers (e.g., lipid measures), brain phenotypes (e.g., grey matter volume), other indexes of health (e.g., diabetes), lifestyles (e.g., physical activity, diet), mental health (e.g., schizophrenia), social variables (e.g., education, income), as well as the cognitive and physical measures used in this work to derive rates of decline. The list of genome-wide summary statistic files used in this work is included in Supplementary Data 6. Two-sample MR was performed using the R-Package *TwoSampleMR*[98,99]. As genetic instruments, we selected LD-independent SNPs (−clump-kb 10,000 −clump-r2 0.001) and MR was performed for all exposures with at least five genetic instruments reaching genome-wide significance ($P < 5 \times 10^{-8}$). The causal effects on the outcome were estimated using inverse-variance weighted (IVW) estimator. To assess the robustness of MR findings, we obtained a number of test diagnostics, including (a) the IVW F-statistic to evaluate the instrument strength (with an F-statistic of > 10 indicating that risk of weak instrument bias is likely to be low[100]), (b) the IVW Q-statistic to assess heterogeneity (with a Q-test $P$-value < 0.05 indicating heterogeneity across instruments[101]), (c) the intercept term in MR-Egger regression to assess possible directional horizontal pleiotropy[102], and (d) MR-PRESSO[103] to assess the robustness against outliers. The summary of MR results includes the standardized causal estimates, the corresponding 95% Confidence intervals and the nominal and Bonferroni-corrected $P$-values, adjusting for the number of exposure dimensions tested (suggestive $P$-value threshold = $P < 0.05/ 11 = 0.005$) and the total number of exposures tested in MR (conservative $P$-value threshold = $P < 0.05/ 106 = 0.0005$). All SNP estimates and corresponding standard errors were standardized as follows prior to inclusion in MR: $\gamma_{STD}(SNP_j) = \frac{\gamma(SNP_j)/SE(SNP_j)}{\sqrt{N_j}}$ and $SE_{STD}(SNP_j) = 1/\sqrt{N_j}$, where $\gamma(SNP_j)$ is the unstandardized effect and $SE(SNP_j)$ the standard error of $SNP j$ on the phenotype and $N_j$ is the sample size per $SNP_j$.

## Reporting summary
Further information on research design is available in the Nature Portfolio Reporting Summary linked to this article.

## Data availability

The genome-wide summary statistics generated as part of this work are deposited in the GWAS Catalogue under accession codes GCST90565836-GCST90565865.

## Code availability

The following software was used to run the analyses:
- REGENIE (https://github.com/rgcgithub/regenie)
- TwoSampleMR (https://mrcieu.github.io/TwoSampleMR/)
- GenomicSEM (https://github.com/GenomicSEM/GenomicSEM).
- otargen (https://amirfeizi.github.io/otargen/)
- LDAK (https://dougspeed.com/).

All analytical scripts are available at https://github.com/TabeaSchoeler/TS2024_UKBBlongitudinal[104].

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

## Acknowledgements

This research has been conducted with the UK Biobank Resource under application number 16389; we thank all biobank participants for sharing their data. We thank all participants involved in the Health Survey England and we thank the Office for National Statistics for granting access to the data. This study would not have possible without the use of publicly available genome-wide summary data and software tools. We acknowledge these resources and thank the research participants, research teams and institutions that have contributed to this research. The computations were performed on the HPC cluster of the Lausanne University Hospital. Z.K. was funded by the Swiss National Science Foundation (# 310030-189147). T.S. was funded by a Wellcome Trust Sir Henry Wellcome fellowship (grant 218641/Z/19/Z). JB.P. has received funding from the European Research Council (ERC) under the European Union's Horizon 2020 research and innovation programme (grant agreement No. 863981).

## Author contributions

Z.K. and T.S. conceptualized the study. T.S. performed the statistical analyses. Z.K., T.S. and J.-B.P. discussed the results and provided comments on the paper. All authors critically reviewed the manuscript.

## Competing interests

The authors declare no competing interests.
