## [Transparent Peer Review file · Nature Communications]

Combining cross-sectional and longitudinal genomic approaches to identify determinants of cognitive and physical decline

Corresponding Author: Dr Tabea Schoeler

Version 0:

Reviewer comments:

Reviewer #1

(Remarks to the Author)

This study is timely, as longitudinal data from large biobanks are increasingly available. It aims to understand the genetic determinants of cognitive and physical declines/aging, compared to baseline levels, by considering three indexes of change and accounting for participation selection using inverse probability weighting. They found the heritability for cognitive or physical aging was negligible compared to that of the corresponding baseline level. Cognitive aging was driven mainly by liability to Alzheimer's disease. In contrast, telomere length and bone mineral density were major drivers for physical aging.

Major

1. Overall, this manuscript is well-written, with proper study design and data analysis considerations. The authors did a great job presenting a framework to guide longitudinal genomic initiatives assuming two-wave data. However, it is unclear how this framework could be adapted for multi-wave data. Could you provide clarification on that?
2. It would help a lot if you could create a figure to conceptualize the framework.
3. Could you provide a rationale for why comparing the genetics underlying cognitive and physical aging is of interest? Additionally, please explain the significance of comparing baseline cognitive/physical function with cognitive/physical aging.
4. Could you create a CONSORT flow diagram to detail the sample selection process for the cross-sectional and longitudinal GWAS, as well as for other analyses (e.g., Figures 2 and 3)? It's unclear whether the same samples were used consistently across all analyses. It seems that some analyses may have included non-European participants.
5. The current manuscript seems lengthy. If it exceeds the word limit, I suggest moving the following sections to the supplement: "Sample Representativeness and Correction for Selective Participation" and "Evaluation of Two-Wave Models of Change for Longitudinal Genetic Effect Estimation."

Minor

1. The files, with a lot of figures, are too big to read on my computer. Please avoid this issue in the next submission.
2. Abstract
 - a. alpha needs to be defined
 - b. No idea why HRH4 on delta is a thing to highlight when I first read the Abstract
3. Methods
 - a. In sFigure 1, the solid line showing the average assessment year across waves is so confusing that I wonder if you can move the information to x-labels.
 - b. Is the age in the equation below age0?
 $\Delta = \text{age} + \text{FU} + \text{age} \times \text{FU} + \text{age}^2$
 - c. Could you justify why you chose $\beta=1$ and $\gamma=0.5$ for all simulations?
 - d. In REGENIE step 2, did you exclude outliers with high missing rate or high heterozygosity based on the field 22027? Can you include the definitions of outliers in the text?
 - e. The Mendelian Randomization (MR) analysis relating 144 exposures to cognitive or physical function/aging only adjusted for 11 exposure dimensions. This approach is liberal, and many of the significant findings are likely false positives. Could

you adjust for the number of exposures per outcome and revisit the interpretation of the results?

f. In the MR paragraph, should $\beta(SE)$ be changed to $\beta(SNP)$?

g. Please report additional MR test results for IVW significant findings to ensure their robustness with respect to MR assumptions. For example, use the IVW F-statistic to evaluate instrument strength, the IVW Q-statistic to assess heterogeneity among causal estimates, the MR-Egger intercept to check for pleiotropy, and MR-PRESSO to assess robustness against outliers.

4. Results

a. The presentation of betas in the Results section is not always clear. Please revise for better clarity.

b. Can you make sure all tables and figures are referenced in the text?

c. Can you clarify how you calculated genetic correlations, using methods like LDSC or simply correlating betas associated with SNPs for a pair of phenotypes (the notations made me think this way, e.g., $r(\beta_{P0}, \beta_{LOG})$)?

(Remarks on code availability)

Reviewer #2

(Remarks to the Author)

The manuscript by Schoeler et al. presents the results from genome-wide association analyses based on multiple longitudinal health-related phenotypes in UK Biobank. The authors used three types of measures per phenotype (absolute, conditional, and relative change) to determine how genetics contributes to the longitudinal changes. They showed that the heritability of these traits is much lower than that of the baseline traits, but they were still able to find genome-wide significant loci for some of them. I really enjoyed reading this manuscript and I think the used approach is innovative and thorough (with the use of different models for change and IPW). Moreover, the presented results are very interesting for the broader scientific community. However, I still have some suggestions that may help to improve the manuscript.

Major comments:

1. It would be good if the authors can also report the results for the genome-wide association studies (and heritability) of the traits obtained at the second wave (i.e. cross-sectional, as they have done for the first wave (baseline)). A comparison of the betas of the two waves will then show if there are any traits where there are loci that only associate at one timepoint, which would be valuable to know.

2. Given that several of the used phenotypes are known to differ between sexes, it would be good if the authors can also include the results of sex-stratified analyses (at least for the genome-wide association analyses).

3. The authors mentioned that the number of individuals with three-wave data was small. However, for some of the phenotypes it still seems large enough (>50,000 cases) to be used for longitudinal analysis. Hence, for those phenotypes it would be interesting if the authors can take all three waves into account in their analyses to see how that influences their results (i.e. use this as a kind of sensitivity analysis). They can then, for example, compare changes between wave 1 and 2 versus those between wave 2 and 3 and wave 1 and 3. It would be valuable to know if data from two timepoints is strong enough to accurately assess longitudinal changes in the UK Biobank and this analysis would be able to partly answer that.

4. It would be good if the authors include a phenotype that does not show any change over time (negative control), to test the validity of their used methods.

Minor comments:

- The authors should provide a reasoning (in the methods section) why they adjusted their models for age². Some readers may not understand why this is relevant.

- I was a bit surprised that the authors included height in their analysis, since the longitudinal changes over time in this phenotype will likely be very minimal. The reported change in Figure 2 is also so small (especially when looking at 3 wave data) that I wonder if this is not just due to fluctuations that are not related to functional decline (e.g. it is known that the time of the day the measurement is taken will have an effect). The authors should discuss this in more detail or consider removing this phenotype from their analyses.

- The authors should check the references to the Supplementary Figures in the main text, they sometimes seem to be wrong.

Joris Deelen

(Remarks on code availability)

Reviewer #3

(Remarks to the Author)

Thank you for the invitation to review the manuscript “Combining cross-sectional and longitudinal genomic approaches to identify determinants of cognitive and physical decline”. The authors present numerous analyses using the UK Biobank data. A lot of effort has gone into this research and the authors used a range of advanced methods, combining cross-sectional and follow-up data, using genome-wide association analysis, heritability estimation and Mendelian randomisation. The authors also compared different study design decisions and tested how these impacted their results. These considerations are important; however, the study lacks focus. Is the aim to test for differences between cross-sectional and ‘longitudinal’ genetic effects? Is this a methodological paper evaluating how best to model data from two time points? Is the aim to identify risk factors for cognitive and physical ‘decline’? Is the aim to assess the representativeness of the UK Biobank follow-up data and to test whether the results remain consistent when applying weighting? These questions are partially addressed in this manuscript, but a common thread is missing. I have included more specific comments below, which the authors may find helpful.

Abstract

It is difficult to view this study as one of aging ‘trajectories’. Cross-sectional data does not allow for exploration of trajectories, and neither do data from two time points.

I was not very familiar with the ‘physical aging’ terminology.

The abstract is lengthy and, as described above, highlights the lack of focus of the study. Only the main finding should be highlighted.

Introduction

Some of the language used is too colloquial (e.g., “poured into research”).

The FEV acronym should be defined when first used.

The distinction between cross-sectional and longitudinal genetic effects is not sufficiently clear and longitudinal genetic effects should be more clearly defined.

The authors comment that prior research has documented the selective participation bias in the UK Biobank. This has also been shown for the follow-up data (e.g., Lyall et al., 2022, doi: 10.1093/braincomms/fcac119).

I suggest the authors refer *differences* in physical and cognitive traits, instead of physical and cognitive aging. For the cross-sectional data by chronological age, even *decline* may not be appropriate, which would suggest changes over time.

Methods

From an epidemiological perspective it is difficult to conceive of analyses of data from two time points as “longitudinal analyses”.

Could the authors provide the rationale for adjusting the change score analyses for follow-up duration?

Please state the hypotheses for the various definitions of change that were analysed.

The analyses of sample representativeness and selective participation in the UK Biobank have in part been performed previously. This could be removed to improve focus of the study.

Perhaps the term “time-invariant” is more appropriate than “cross-sectional” in distinguishing longitudinal genetic effects, i.e., those expected to differ throughout the lifespan.

Results

A limitation of describing the data presented in Figure 2 as decline is that these are cross-sectional data and differences could be due to alternative explanations (e.g., cohort effects).

The finding that baseline-adjustment likely results in bias is of interest. How does this compare to the clinical trial literature?

Discussion

Revise “male gender” to “male sex”.

The authors use the term ‘accelerated aging’; however, it is not clearly defined in the context of this work.

The conclusion that a higher baseline function in a given trait or factors shaping that trait may not serve as a buffer against age-related decline contradicts the concept of ‘cognitive reserve’. Please discuss.

Given the final paragraph highlighting the benefits of the analytical approach, what additional preventative strategies were identified using this approach?

Notably a limitation section is missing and the discussion of findings from other studies is minimal.

(Remarks on code availability)

Version 1:

Reviewer comments:

Reviewer #1

(Remarks to the Author)

Thank you for the revisions and updates made to your manuscript. I appreciate your thorough attention to my comments and suggestions. After reviewing the changes, I confirm that my queries have been adequately addressed, and I recommend this manuscript for publication.

(Remarks on code availability)

Reviewer #2

(Remarks to the Author)

The authors have done an excellent job in revising the manuscript based on my and the other reviewers' suggestions. However, I think they misunderstood my first comment. My point was not that they should use the "comparison of two cross-sectional GWA as a mean to estimate age-varying genetic effects", but I was interested to see if a cross-sectional GWAS done in the same individuals at different timepoints results in different associations, i.e. how much do these cross-sectional associations fluctuate. One could use this to see how replicable cross-sectional associations are. However, this was more a question that was driven by my own interests and I do not feel there is a need for the authors to address this to improve manuscript.

(Remarks on code availability)

I have reviewed the code and the README file provides enough instructions about the used scripts (which have also been provided) and R packages. However, I do not have enough bioinformatic expertise to judge if they have used the best possible methods for their analyses.

Reviewer #3

(Remarks to the Author)

Review of NCOMMS-24-62971A

Thank you for the opportunity to see the revised version of this manuscript. The authors have sufficiently addressed my comments, and I have no further concerns. I congratulate the authors for their efforts and look forward to seeing the published version of this manuscript.

(Remarks on code availability)

We thank the reviewers for their thorough review and constructive feedback on our work. In light of the comments, we have made three main changes, including 1) shortening of the manuscript, 2) additional sensitivity analyses (exploration of 3-wave models of change, test of Mendelian Randomization assumptions, tests of sex-dependent genetic effects), and 3) extended discussions on the theoretical background and existing evidence focusing on aetiological questions in the context of age-related decline.

Reviewer 1

This study is timely, as longitudinal data from large biobanks are increasingly available. It aims to understand the genetic determinants of cognitive and physical declines/aging, compared to baseline levels, by considering three indexes of change and accounting for participation selection using inverse probability weighting. They found the heritability for cognitive or physical aging was negligible compared to that of the corresponding baseline level. Cognitive aging was driven mainly by liability to Alzheimer's disease. In contrast, telomere length and bone mineral density were major drivers for physical aging.

Major comments:

R1.1 Overall, this manuscript is well-written, with proper study design and data analysis considerations. The authors did a great job presenting a framework to guide longitudinal genomic initiatives assuming two-wave data. However, it is unclear how this framework could be adapted for multi-wave data. Could you provide clarification on that?

We thank the reviewer for highlighting the usefulness of our framework for longitudinal genomic work using two-wave data. When using three-wave data (or more), linear slopes of change can be obtained from mixed effect models including a random slope and intercept (describing the overall level and change across time, respectively), where the longitudinal phenotype is expressed as a simple linear function of time. The slope obtained from that model would approximate the slope obtained from our two-wave models, albeit increased precision resulting from the addition of an extra measurement occasion. Thus, all our analysis pipeline can be directly adapted to data with more than two time points.

For reasons discussed in the paper (e.g., data availability, sample representativeness), we focused specifically on two-wave instead three-wave models. Nevertheless, we have explored possible gains obtained from using three-wave data (in response to **R2.3**), which also illustrates how our framework can be adapted once additional assessment waves become available.

[Manuscript, Methods]: "Of note, as three-wave data for fluid intelligence was available for a relatively large subset (69,801 participants), we explored possible gains resulting from adding one more time point (c.f., sDiscussion, Supplement)."

[Supplement, sDiscussion]: "Slopes obtained from three-wave data: To evaluate whether adding an additional time point improves the measurement of change, we derived slopes of change for one phenotype (fluid intelligence) with at least

three non-missing observations ($N = 69,801$). First, we applied a linear mixed model (LMM) with random intercepts and slopes to estimate individual slopes of change. However, the LMM resulted in singular fits, suggesting that the model structure was too complex for the data. We therefore switched to ordinary least squares (OLS) regression. For each individual with three-wave data, we regressed the phenotype (Y_t) at time point t on the years elapsed since baseline ($t = 0$) using the model $Y_t \sim \text{years}_t$. The individual regression coefficients were then taken as the three-wave slopes of change.

We then compared the three-wave slopes (Δ_{W3}) to two-wave slopes of change ($\Delta_{W2} = \frac{P_1 - P_0}{FU}$), which were derived based on the difference between the phenotype assessed at baseline (P_0) and the most recent follow-up assessment (P_1), divided by the follow-up time (FU). Comparing the two slopes of change (Δ_{W2} and Δ_{W3}) revealed a high correlation between the two ($r = 0.96$), and the addition of one more time point increased the between-subject variance from 41.4% to 48.2%, indicating improved precision in Δ_{W3} . Specifically, the total variance of the phenotype is composed of $Var(P) = \sigma_I^2 + \sigma_V^2 + \sigma^2$, where σ_I^2 corresponds to the time-invariant, between subject variance (e.g., genetic), σ_V^2 to the time-variant (within-subject) change and σ^2 to the measurement error. Incorporating additional time points would reduce measurement error (σ^2), thereby increasing the proportion of the variance explained by between-subject differences ($\frac{\sigma_I^2}{\sigma_I^2 + \sigma_V^2 + \sigma^2}$).

In summary, the additional time point modestly improved the precision of the slope of change ($\sim 6.8\%$), which has implications for the estimation of variance components (e.g., h^2 of change) and statistical power to detect age-varying genetic effects. However, the larger sample size available from two-wave data for fluid intelligence (159,762), along with greater sample representativeness, likely outweighs the modest precision gains achieved by the incorporation of the additional time point.”

R1.2 It would help a lot if you could create a figure to conceptualize the framework.

We have now included the following figure to illustrate the study framework:

[Manuscript, Methods]: “The methodological framework of this work is illustrated in Figure 1.”

Figure 1. Study framework

Analytical framework to study longitudinal genetic effects on phenotypes subject to age-related decline. The structural causal model is shown at the top, where the phenotype P at time point t (P_0 = baseline, P_1 = follow-up) varies as a function of time-invariant (baseline) genetics (G_0), time-varying (longitudinal) genetics (G_E) and the environment (E). Simulations are used (middle panel) to assess the suitability of three definitions of change (Δ) for longitudinal genome-wide analyses ($\Delta \sim \gamma \cdot G$), including absolute change (Δ_{DIFF} , i.e., the absolute difference between the baseline phenotype, P_0 , and the follow-up phenotype, P_1), relative change (Δ_{LOG} , i.e., the difference between $\log(P_0)$ and $\log(P_1)$) and conditional change (Δ_{RES} , i.e., the difference between the observed P_1 and the predicted P_1 phenotype). Genome-wide tests and downstream analyses (bottom panel) are performed on composite scores of cross-sectional (i.e., time-invariant) and longitudinal (i.e., time-varying) indexes of cognitive and physical aging.

R1.3 Could you provide a rationale for why comparing the genetics underlying cognitive and physical aging is of interest? Additionally, please explain the significance of comparing baseline cognitive/physical function with cognitive/physical aging.

We have now provided a more in-depth rationale for the comparison of cognitive and physical dimensions and the distinction between lifetime genetic effects (i.e., cross-sectional) and age-varying genetic effects (i.e., longitudinal):

[Manuscript, Introduction]: [“... For that purpose, the UK Biobank (UKBB) has curated and released a rich set of prospectively ascertained aging phenotypes, capturing key aspects of changes in physical (e.g., FEV, fitness levels, grip strength) and cognitive (e.g., reaction time, fluid intelligence) dimensions. This resource now enables the investigation of prominent aging theories within a genetically informed framework, including common cause theories of aging¹⁸⁻²¹ or the role cognitive/physical reserve in age-related decline²²⁻²⁸ (c.f., ‘Research in context’ in the Supplement for further discussion).”

References:

- ¹⁸Clouston, S. A. et al. The dynamic relationship between physical function and cognition in longitudinal aging cohorts. *Epidemiologic reviews* 35, 33–50 (2013).
- ¹⁹Christensen, H. et al. Are changes in sensory disability, reaction time, and grip strength associated with changes in memory and crystallized intelligence? A longitudinal analysis in an elderly community sample. *Gerontology* 46, 276–292 (2000).
- ²⁰Baltes, P. B. & Lindenberger, U. Emergence of a powerful connection between sensory and cognitive functions across the adult life span: A new window to the study of cognitive aging? *Psychology and aging* 12, 12 (1997).
- ²¹Zammit, A. R., Robitaille, A., Piccinin, A. M., Muniz-Terrera, G. & Hofer, S. M. Associations between aging-related changes in grip strength and cognitive function in older adults: A systematic review. *The Journals of Gerontology: Series A* 74, 519–527 (2019).
- ²²Gow, A. J. et al. Is age kinder to the initially more able?: Yes, and no. *Intelligence* 40, 49–59 (2012).
- ²³Christensen, H. & Henderson, A. Is age kinder to the initially more able? A study of eminent scientists and academics. *Psychological medicine* 21, 935–946 (1991).
- ²⁴Deary, I. J., Starr, J. M. & MacLennan, W. J. Is age kinder to the initially more able?: Differential ageing of a verbal ability in the healthy old people in Edinburgh study. *Intelligence* 26, 357–375 (1998).
- ²⁵Stern, Y. What is cognitive reserve? Theory and research application of the reserve concept. *Journal of the international neuropsychological society* 8, 448–460 (2002).
- ²⁶Stern, Y. et al. Whitepaper: Defining and investigating cognitive reserve, brain reserve, and brain maintenance. *Alzheimer’s & Dementia* 16, 1305–1311 (2020).
- ²⁷Kuh, D., Karunanathan, S., Bergman, H. & Cooper, R. A life-course approach to healthy ageing: Maintaining physical capability. *Proceedings of the Nutrition Society* 73, 237–248 (2014).
- ²⁸Salthouse, T. A. Mental exercise and mental aging: Evaluating the validity of the ‘use it or lose it’ hypothesis. *Perspectives on psychological science* 1, 68–87 (2006).

[Supplement, ‘Research in context’]: “Studying age-varying genetic effects is important for a number of reasons. First, such work allows to assess if the genetic architecture underlying lifetime levels of function is similar to that characterizing age-related change. In other words, while tests on change quantify the impact of age on genome-wide associations, cross-sectional genome-wide analyses identify marginal genetic effects that are assumed to be constant over time (c.f., study framework, **Figure 1** in the main manuscript). Second, genetically informed designs allow to further probe existing theoretical models of aging. For example, one key interest concerns the question as to whether ‘age is kinder to the initially more able’³⁰⁻³², as innate and/or early life resources may slow down age-related decline. Similar prediction are made based on theories of ‘cognitive reserve’³³, ‘brain reserve’³⁴, ‘health capital’³⁵ or ‘differential preservation’³⁶, broadly hypothesizing that individuals with higher levels of reserve are more protected from neurobiological decay throughout the aging process. Evidence on this topic has, however, largely been mixed: While some studies suggest that health

reserves may protect against age-related decline, including evidence linking educational attainment to slower cognitive decline³⁷⁻⁴⁰, a larger body of research has failed to replicate these findings⁴¹⁻⁴⁹. Our findings are consistent with the latter body of research, suggesting that high baseline function in a given trait (e.g., cognition) or related factors (e.g., educational attainment) may not buffer against age-related decline in that trait. In other words, while cognitive or physical reserves can delay the onset of functional impairment, they are unlikely to slow the overall rate of decline. Other lines of research have focused on common cause theories of aging, investigating whether age-related declines in physical and cognitive domains are driven by shared aetiological processes⁵⁰⁻⁵³. While small but significant associations between cognitive and physical decline have indeed been documented⁵⁰, findings remains largely inconsistent^{43,50,53-56}. In our work, we observed some indications of shared risks (e.g., shorter parental lifespan linking to both increased cognitive and physical decline) when tested in Mendelian Randomization analyses. However, most implicated risk factors demonstrated independent effects across the physical and cognitive dimensions of decline.”

R1.4 Could you create a CONSORT flow diagram to detail the sample selection process for the cross-sectional and longitudinal GWAS, as well as for other analyses (e.g., Figures 2 and 3)? It's unclear whether the same samples were used consistently across all analyses. It seems that some analyses may have included non-European participants.

We thank the reviewer for this suggestion and agree that a flow chart could help the reader to better understand at which stages participants were excluded from the analyses (e.g., non-European samples were included in phenotypic analyses but not genome-wide analyses). Since the template is designed for parallel group randomised trials, we have slightly amended the design to incorporate the non-randomized design of the UK Biobank and the different types of analyses (i.e., phenotypic versus genome-wide analyses):

[Manuscript, Methods]: “More details regarding the sample selection process for the cross-sectional and longitudinal analyses are provided in **sFigure 4** (Supplement).”

sFigure 4. Flow Diagram of the sample selection process

R1.5 The current manuscript seems lengthy. If it exceeds the word limit, I suggest moving the following sections to the supplement: "Sample Representativeness and Correction for Selective Participation" and "Evaluation of Two-Wave Models of Change for Longitudinal Genetic Effect Estimation."

To reduce the word limit of the manuscript, we have now moved the sections "Sample Representativeness and Correction for Selective Participation" and "Evaluation of Two-Wave Models of Change for Longitudinal Genetic Effect Estimation" from the Method/Result section to the Supplement.

Minor comments:

R1.6. The files, with a lot of figures, are too big to read on my computer. Please avoid this issue in the next submission.

We have now split some of the multi-panel figures to enhance the readability.

R1.7. alpha needs to be defined

This is now defined in the abstract (“cognitive decline was largely driven by Alzheimer’s disease liability (standardized MR-effect, $\gamma = 0.17$). Of note, α was replaced by γ in response to R1.14”

R1.8. No idea why HRH4 on delta is a thing to highlight when I first read the Abstract

We have removed some of the finding relating to participation bias correction in the abstract (in response to R3.3), as part of which we also removed the mentioning of this specific finding.

R1.9. In sFigure 1, the solid line showing the average assessment year across waves is so confusing that I wonder if you can move the information to x-labels.

We have now removed the solid line highlighting the average assessment year and included this information in the x-axis labels instead (c.f., example below).

Amended legend text: “Selected aging phenotypes used to derive indexes of physical and cognitive decline. The x-axis specifies the environment in which the phenotype was assessed at a given time point (i.e., assessment centre visit or online), the number of waves available for the phenotype and the mean year when the assessment took place per wave.”

R1.10. Is the age in the equation below age0?

$$\Delta = \text{age} + FU + \text{age} \times FU + \text{age}^2$$

Indeed, we have now added this in the manuscript ($\Delta = \text{age}_0 + FU + \text{age}_0 \times FU + \text{age}_0^2$).

R1.11. Could you justify why you chose beta=1 and gamma=0.5 for all simulations?

These parameter specifications were chosen to indicate greater environmental contribution (β) than gene-by-environment (γ) or genetic (α) contribution to the traits. While changing these parameters will impact a number of diagnostics we used when evaluating risk of bias resulting from the different definitions of change, the overall conclusions are not altered. For example, when simulating data based on the parameters quoted above ($\beta=1, \gamma=0.5, \alpha=0.1$), we found a false positive rate of 22% when using residual change scores (Δ_{RES}). As illustrated in the Figure below, decreasing β and γ (i.e., the environmental part) while keeping the genetic effect constant ($\beta=0.5, \gamma=0.25, \alpha=0.1$, c.f., Simulation 1 below) increases the false positive rates from 22% to 67% when using Δ_{RES} . This is because the genetic effects are gaining importance, thereby amplifying the bias induced by baseline genetic effects when using Δ_{RES} . In contrast, increasing the environmental contribution ($\beta=2, \gamma=1, \alpha=0.1$, c.f., Simulation 2 below) reduces the false positive rate when using Δ_{RES} (from 22% to 9%) as the genetic component loses importance. As such, while different parameter specifications impact the degree of bias resulting from the different definitions of change, the overall conclusions drawn from the simulations remain (that it, baseline adjustment increases risk of false positive findings and the stronger the baseline genetic effect is, the larger the bias will be).

● Δ_{LOG} ● Δ_{DIFF} ● Δ_{RES}

Simulation 1. $\beta=0.5, \gamma=0.25, \alpha=0.1$

Simulation 2. $\beta=2, \gamma=1, \alpha=0.1$

R1.12. In REGENIE step 2, did you exclude outliers with high missing rate or high heterozygosity based on the field 22027? Can you include the definitions of outliers in the text?

Outliers with high missing rate or high heterozygosity were excluded, which we had specified in the section ‘Longitudinal and cross-sectional genome-wide scans and downstream analyses’ of the initial manuscript (c.f., “The sample was restricted to individuals of European genetic ancestry and individuals with high missing rate and/or high heterozygosity on autosomes (as determined by the UKBB³⁸) were excluded.”). To make this clearer to the reader, we have now also included this information in the Flow Diagram of the sample selection process (c.f., sFigure 4, shown in response to **R1.4**).

R1.13 Mendelian Randomization

- a. The Mendelian Randomization (MR) analysis relating 144 exposures to cognitive or physical function/aging only adjusted for 11 exposure dimensions. This approach is liberal, and many of the significant findings are likely false positives. Could you adjust for the number of exposures per outcome and revisit the interpretation of the results?

We agree that the adjustment for multiple testing based on 11 exposure dimensions is liberal, which we had chosen to incorporate the correlated nature of many of the included exposures (e.g., self-reported childhood BMI and measured childhood BMI) and to reduce the risk of false-negatives. However, we agree with the reviewer’s remark and have now highlighted the findings following conservative Bonferroni correction. Of note, while the reviewer refers to 144 exposures, this number refers to the number of genome-wide summary statistic files initially selected for MR analysis (c.f., Methods: “We included data from 144 genome-wide studies to extract the genetic instruments”). Since we only included exposures with at least 5 genetic instruments [Methods: “MR was performed for all exposures with at least five genetic instruments reaching genome-wide significance ($P < 5 \times 10^{-8}$)], the number of exposures tested in MR was less slightly less (105 instead of 144). We have now revised the manuscript as follows:

[Manuscript, Methods]: “The summary of MR results includes the standardized causal estimates, the corresponding 95% Confidence intervals and the nominal and Bonferroni-corrected P -values, adjusting for the number of exposure dimensions tested (suggestive P -value threshold = $P < 0.05/ 11= 0.005$) and the total number of exposures tested in MR (conservative P -value threshold = $P < 0.05/ 105=0.0005$).”

[Manuscript, Results]: “For example, while cognitive decline was mostly predicted (at nominal significance level) by Alzheimer’s disease, lipid traits (e.g., Apolipoprotein A and B) and behaviours potentially altering those (e.g., vegetable intake), there was little overlap with risks identified for physical decline (e.g., shorter telomere length, higher bone mineral density, basal metabolic rate, poor sleep). 10 exposures on cognitive/physical decline survived suggestive Bonferroni correction for multiple testing ($P < 0.05/ 11= 0.005$, were 11 is the number of independent exposure dimensions) (c.f., triangle shapes in **Figure 5**), of which 7 (highlighted in rectangular shapes) remained significant following stringent Bonferroni correction ($P < 0.05/ 105= 0.0005$, were 105 is

the number of exposures tested in MR). Of note, most of the stringently identified risk factors pointed towards discrepancies resulting from scale-dependencies. For example, higher baseline physical function predicted steeper physical decline when assessed in absolute ($\gamma = 0.1$, $P = 8.1e-14$) but not relative ($\gamma = 0.02$, $P = 0.16$) terms, a pattern consistent with the presence of non-linear change.”

Header of updated Figure 5:

Updated Legend: Standardized Mendelian Randomization effect estimates (β_{STD}) of exposure effects on cross-sectional outcomes (P_0 in red, with positive coefficients indexing higher levels of function) and longitudinal outcomes (Δ_{DIFF} in blue and Δ_{LOG} in orange, with positive coefficients indexing larger decline). Filled points (\bullet), triangles (\blacktriangle) and diamonds (\blacklozenge) highlight significant effects (at $P < 0.05$, $P < 0.05/11$ and $P < 0.05/105$, respectively). Circles (\circ) highlight non-significant ($P > 0.05$) effects.

- b. In the MR paragraph, should beta(SE) be changed to beta(SNP)
The typo has been corrected in the revised version of the manuscript.
- c. Please report additional MR test results for IVW significant findings to ensure their robustness with respect to MR assumptions. For example, use the IVW F-statistic to evaluate instrument strength, the IVW Q-statistic to assess heterogeneity among causal estimates, the MR-Egger intercept to check for pleiotropy, and MR-PRESSO to assess robustness against outliers.

We have now performed the suggested sensitivity analyses for MR and included the results in the Mendelian Randomization Supplement Table. Globally, the initial conclusions were confirmed by more robust MR methods/tests. We discussed the findings in the manuscript as follows:

[Manuscript, Methods]: “The causal effects on the outcome were estimated using inverse-variance weighted (IVW) estimator. To assess the robustness of MR findings, we obtained a number of test diagnostics, including (a) the IVW F-statistic to evaluate the instrument strength (with an F-statistic of >10 indicating that risk of weak instrument bias is likely to be low⁵⁸), (b) the IVW Q-statistic to assess heterogeneity (with a Q-test P -value < 0.05 indicating heterogeneity across instruments⁵⁹), (c) the intercept term in MR-Egger regression to assess possible directional horizontal pleiotropy⁶⁰, and (d) MR-PRESSO⁶¹ to assess the robustness against outliers.

[Manuscript, Results]: “Further, parental lifespan (PLS) on cognitive decline was flagged as possibly biased based on a number of MR sensitivity analyses, given the level of heterogeneity (Q-test $P = 3.2e-08$), the significant MR-Egger intercept term ($P = 1.5e-05$ for $PLS \rightarrow \Delta_{LOG}$) and the non-significant MR-PRESSO effects (e.g., $P = 0.23$). The complete set of MR-results, including the exposure-outcome associations when performing MR on the individual aging phenotypes, is included in sTable 8 and sFigure 14/15 (Supplement).”

R1.14 Results

- a. The presentation of betas in the Results section is not always clear. Please revise for better clarity

We have now revised the result section and corresponding figures and used ' α ' when referring to age effects (first part in the result section), ' β ' when referring to genetic variant effects (second part in the result section) and ' γ ' when referring to MR effect estimates (third part in the result section).

- b. Can you make sure all tables and figures are referenced in the text?

We now referenced all tables and figures in the main text.

- c. Can you clarify how you calculated genetic correlations, using methods like LDSC or simply correlating betas associated with SNPs for a pair of phenotypes (the notations made me think this way, e.g., $r(\beta_{P0}, \beta_{LOG})$)?

We assume the reviewer refers to the dashed line shown in Figure 4 in the main manuscript (example shown below), indexing the association between the cross-sectional and longitudinal SNP effects. The slope was obtained from a linear regression model, regressing the SNP effects on change onto the SNP effects on the baseline phenotype. We have now made this clearer in the legend of that figure ("The dashed slope (line of best fit, obtained from $\beta_{\Delta} \sim \beta_{P0}$) represents the association between the cross-sectional and longitudinal SNP effects.")

If the reviewer refers to the results described in MR result section (testing the relationship between MR-effects on baseline function and longitudinal change), these were obtained from Pearson correlation tests. As before, we have now made this more explicit in the manuscript:

[Manuscript, Results]: "Further, we found little evidence of common underpinnings shared by (cross-sectional) level of functioning and (longitudinal) decline in cognitive and physical outcomes, implicated by the lack of association between cross-sectional MR-effects (γ_{P0}) and longitudinal MR-effects when modelled as relative change [Pearson correlation coefficient $r(\gamma_{P0}, \gamma_{LOG}) = -0.06$, $P = 0.56$] or absolute change [$r(\gamma_{P0}, \gamma_{DIFF}) = 0.03$, $P = 0.72$]."

Reviewer 2

The manuscript by Schoeler et al. presents the results from genome-wide association analyses based on multiple longitudinal health-related phenotypes in UK Biobank. The authors used three types of measures per phenotype (absolute, conditional, and relative change) to determine how genetics contributes to the longitudinal changes. They showed that the heritability of these traits is much lower than that of the baseline traits, but they were still able to find genome-wide significant loci for some of them. I really enjoyed reading this manuscript and I think the used approach is innovative and thorough (with the use of different models for change and IPW). Moreover, the presented results are very interesting for the broader scientific community. However, I still have some suggestions that may help to improve the manuscript.

We thank the reviewer for this positive feedback on our work.

Major comments:

R2.1 It would be good if the authors can also report the results for the genome-wide association studies (and heritability) of the traits obtained at the second wave (i.e. cross-sectional, as they have done for the first wave (baseline)). A comparison of the betas of the two waves will then show if there are any traits where there are loci that only associate at one timepoint, which would be valuable to know.

We appreciate the reviewer's suggestion to compare results obtained from two cross-sectional genome-wide tests, performed separately on 1) the baseline phenotype and 2) the follow-up phenotype. However, inferring longitudinal genetic effects from comparisons of two cross-sectional GWA (instead of directly estimating them in GWA on change) is unlikely to yield meaningful insights, as can be seen in the following:

As described in the manuscript, the longitudinal (i.e., age-dependent) effects are conceptualized as gene \times environment interactions, where the observed phenotype P at time point t is modelled as a function of time-invariant genetics (G_0), time-varying (longitudinal) genetics (G_E) and the environment (E): $P_t = \alpha \cdot G_0 + \beta \cdot E_t + \gamma \cdot G_E \cdot E_t + \varepsilon_t$. Following the reviewer's suggestion, we would estimate the (time-varying) genetic effects ($\gamma \cdot G_E$) from a comparison of cross-sectional GWA on the baseline and follow-up phenotype:

Baseline association tests: $P_0 = \alpha \cdot G_0 + \beta \cdot E_0 + \gamma \cdot G_E \cdot E_0 + \varepsilon_0$

Follow-up association tests: $P_1 = \alpha \cdot G_0 + \beta \cdot E_1 + \gamma \cdot G_E \cdot E_1 + \varepsilon_1$

As the time-invariant genetic effects ($\alpha \cdot G_0$) cancel each other out, the difference between P_1 and P_0 can be written as: $P_1 - P_0 = \beta \cdot (E_0 - E_1) + \gamma \cdot G_E \cdot (E_1 - E_0) + (\varepsilon_1 - \varepsilon_0)$. As such, a comparison at the cross-sectional level is equivalent to our model using the longitudinal data structure, but would likely result in a loss of statistical power. However, more problematically, identifying 'loci that only associate at one timepoint' would essentially mean to infer time-varying genetic effects from an interpretation null findings [i.e., time-varying genetic effects if $P(\gamma_{P_0}) > 5 \times 10^{-8}$ and $P(\gamma_{P_1}) < 5 \times 10^{-8}$ OR if $P(\gamma_{P_0}) < 5 \times 10^{-8}$ and $P(\gamma_{P_1}) > 5 \times 10^{-8}$], which is not a valid approach for the identification of age-varying effects as significance at one time point,

but not at another one can simply be due to random fluctuations or differences in the available sample size. Given these limitations, we decided to abstain from a comparison of two cross-sectional GWA as a mean to estimate age-varying genetic effects.

R2.2 Given that several of the used phenotypes are known to differ between sexes, it would be good if the authors can also include the results of sex-stratified analyses (at least for the genome-wide association analyses).

We thank the reviewer for pointing out possible sex-dependent effects. To explore this question further, we have now performed genotype-by-sex tests on the cognitive and physical composite scores in both cross-sectional and longitudinal GWAs. We have updated the method and result section accordingly, which now reads as follows:

[Methods, Manuscript]: “As age-related change in cognitive and physical function have been reported to differ between sexes^{51,52}, we also estimated sex-dependent genome-wide effects. For that, we extended our models on change and baseline level of function by including a gene-by-sex interaction term ($\Delta = \beta_1 \cdot G + \beta_2 \cdot G \cdot SEX + \varepsilon$ and $P_0 = \beta_1 \cdot G + \beta_2 \cdot G \cdot SEX + \varepsilon$, respectively).”

References:

⁵¹Okabe, T. et al. Sex differences in age-related physical changes among community-dwelling adults. *Journal of Clinical Medicine* 10, 4800 (2021).

⁵²McCarrey, A. C., An, Y., Kitner-Triolo, M. H., Ferrucci, L. & Resnick, S. M. Sex differences in cognitive trajectories in clinically normal older adults. *Psychology and aging* 31, 166 (2016).

[Results, Manuscript]: “Among the six phenotypes assessed in interaction testing (cross-sectional physical and cognitive function, absolute cognitive and physical decline, and relative cognitive and physical decline), two genetic variants exhibited sex-specific effects: rs13141641 (nearest gene: *HHIP*) and rs9748016 (*RFLNB*), both of which were identified in the genome-wide tests on baseline physical function. Mapping these variants to prior phenotype-genotype associations revealed that these sex-dependent genetic effects mostly linked to indices of physical health, such as lung function and heel bone mineral density (**sFigure 14**).”

sFigure 17. Genotype-phenotype associations of variants showing sex-differential effects on physical function (composite score)

Figure 17: SNPs showing significant gene-by-sex interaction effects ($P < 5 \times 10^{-8}$) were annotated to the nearest gene and mapped to previously associated phenotypes using the Open Target Genetics database, which curates summary statistic files from four sources: 1) GWAS analyses by NEALE (<http://www.nealelab.is/uk-biobank>) in the UK Biobank, 2) SAIGE analyses on binary phenotypes in the UKBB (<https://www.leelabsg.org/resources>), 3) the GWAS Catalog (GCST) (<https://www.ebi.ac.uk/gwas/>) and 4) FinnGen (<https://www.fingene.fi/>). The horizontal line in black represents the significant threshold ($P < 5 \times 10^{-8}$)

R2.3 The authors mentioned that the number of individuals with three-wave data was small. However, for some of the phenotypes it still seems large enough (>50,000 cases) to be used for longitudinal analysis. Hence, for those phenotypes it would be interesting if the authors can take all three waves into account in their analyses to see how that influences their results (i.e. use this as a kind of sensitivity analysis). They can then, for example, compare changes between wave 1 and 2 versus those between wave 2 and 3 and wave 1 and 3. It would be valuable to know if data from two timepoints is strong enough to accurately assess longitudinal changes in the UK Biobank and this analysis would be able to partly answer that.

We appreciate the reviewer’s comment, which points to the important question concerning the trade-off between utilizing two-wave change (which is available in larger N and is less affected by participation bias, but may reduce the precision of the slopes of change) and using three or more waves, which can offer greater precision but is often only available in smaller (less representative) samples. To explore this question, we focused on ‘fluid intelligence’ as this measure provided the largest N for three-wave data ($N \sim 69,000$). While the inclusion of an additional time point did indeed increase the precision of the slope of change, we found that the gains were rather moderate and may not outweigh the benefits from two-wave data. Purely from the statistical standpoint, moving from two to three observations reduces the squared SE of the slope estimate by one third, but the sample size

would be reduced by much more than one third. Therefore, the overall precision is still better with two time points at the current data availability pattern. We have now discussed the findings and implications in more detail in the Supplement:

[Supplement, sDiscussion]: “**Slopes obtained from three-wave data:** To evaluate whether adding an additional time point improves the measurement of change, we derived slopes of change for one phenotype (fluid intelligence) with at least three non-missing observations ($N = 69,801$). First, we applied a linear mixed model (LMM) with random intercepts and slopes to estimate individual slopes of change. However, the LMM resulted in singular fits, suggesting that the model structure was too complex for the data. We therefore switched to ordinary least squares (OLS) regression. For each individual with three-wave data, we regressed the phenotype (Y_t) at time point t on the years elapsed since baseline ($t = 0$) using the model $Y_t \sim \text{years}_t$. The individual regression coefficients were then taken as the three-wave slopes of change.

We then compared the three-wave slopes (Δ_{W3}) to two-wave slopes of change ($\Delta_{W2} = \frac{P_1 - P_0}{FU}$), which were derived based on the difference between the phenotype assessed at baseline (P_0) and the most recent follow-up assessment (P_1), divided by the follow-up time (FU). Comparing the two slopes of change (Δ_{W2} and Δ_{W3}) revealed a high correlation between the two ($r = 0.96$), and the addition of one more time point increased the between-subject variance from 41.4% to 48.2%, indicating improved precision in Δ_{W3} . Specifically, the total variance of the phenotype is composed of $Var(P) = \sigma_I^2 + \sigma_V^2 + \sigma^2$, where σ_I^2 corresponds to the time-invariant, between subject variance (e.g., genetic), σ_V^2 to the time-variant (within-subject) change and σ^2 to the measurement error. Incorporating additional time points would reduce measurement error (σ^2), thereby increasing the proportion of the variance explained by between-subject differences ($\frac{\sigma_I^2}{\sigma_I^2 + \sigma_V^2 + \sigma^2}$).

In summary, the additional time point modestly improved the precision of the slope of change ($\sim 6.8\%$), which has implications for the estimation of variance components (e.g., h^2 of change) and statistical power to detect age-varying genetic effects. However, the larger sample size available from two-wave data for fluid intelligence (159,762), along with greater sample representativeness, likely outweighs the modest precision gains achieved by the incorporation of the additional time point.”

Comparing changes between wave 1 and 2 versus those between wave 2 and 3 and wave 1 and 3, as suggested by the reviewer, has suboptimal power for detecting change. More specifically, for individuals with more than two waves of assessments, we had used data from the most recent assessment as the follow-up point, to maximize the follow-up duration and increase power for the detection of aging-varying genetic effects. For example, for an individual assessed in 2012, 2017 and 2022, we would have used the phenotype difference between 2012 and 2022, thereby making use of the total follow-up duration (i.e., 10 years).

Using wave 1-2 change and wave 2-3 change instead would therefore substantially reduce the follow-up duration (to 5 years instead of 10), thereby minimizing power for the detection of factors associated with change. On the other hand, contrasting changes between wave 1 and 2 versus those between wave 2 and 3 could be used to detect non-linear change patterns. While this is an interesting question, due to very low power, we would leave it for later research when larger samples with three non-missing observations become available.

R2.4 It would be good if the authors include a phenotype that does not show any change over time (negative control), to test the validity of their used methods.

We thank the reviewer for suggesting the use of negative controls. While we agree that such approach represents a valuable sensitivity analysis, we believe that a valid negative control cannot be derived from the data at hand, for two main reasons: First (1), we included only objectively ascertained aging phenotypes to avoid problems resulting from age-related measurement error in self-report measures (e.g., memory problems affecting the accuracy of self-reported memory problems itself). As there are no time-invariant phenotypes available from objective measures of health in the UK Biobank, we are unable to select a negative control that is comparable to our selected aging phenotypes. Alternatively (2), time-invariant phenotypes obtained from self-report measures could potentially serve as a negative control (e.g., self-reported birth weight assessed at baseline and at follow-up), as there should be no genetic effects on change. While this idea seems intuitive, we had previously shown that variations in self-report inconsistencies over time (including self-reported birth weight) are not random and can be predicted based on genetic data (c.f., reference below, Schoeler et al., 2024). For example, self-report inconsistencies over time showed significant SNP-heritability (~3%) and were predicted by a number of traits, such as age, sex or educational attainment. Considering that these outcomes do not serve as a valid negative control, we decided to abstain from analyses on self-report inconsistencies over time.

Reference: Schoeler, T., Pingault, J. B., & Kotalik, Z. (2024). The impact of self-report inaccuracy in the UK Biobank and its interplay with selective participation. *Nature Human Behaviour*, 1-11.

Minor comments:

R2.5 The authors should provide a reasoning (in the methods section) why they adjusted their models for age². Some readers may not understand why this is relevant.

To clarify the inclusion of age² in our models, we have included the following:

[Manuscript, Methods]: “Age₀² was included in our models to account for possible non-linear rates of change, i.e., situations where the effect of age on within-individual change increases over time.”

R2.6 I was a bit surprised that the authors included height in their analysis, since the longitudinal changes over time in this phenotype will likely be very minimal. The reported

change in Figure 2 is also so small (especially when looking at 3 wave data) that I wonder if this is not just due to fluctuations that are not related to functional decline (e.g. it is known that the time of the day the measurement is taken will have an effect). The authors should discuss this in more detail or consider removing this phenotype from their analyses.

We agree that height shows less variability over time than other phenotypes (e.g., Forced Expiratory Volume, FEV). While this could indicate that there is less true change for height than FEV as individuals age, it could also indicate that height is more accurately measured and therefore less prone to measurement error than other phenotypes included in this work. Interestingly, GWA analysis of change in height revealed that this phenotype exhibited one of the highest levels of SNP heritability ($h^2 \sim 2\%$) among all indexes of change examined in this work. As such, we believe that change in height represents a valuable measure for GWA on age-related decline, as it can relate to muscle loss (around the spine) that would diminish posture and hence result in shorter stature.

R2.7 The authors should check the references to the Supplementary Figures in the main text, they sometimes seem to be wrong.

We have now revised the references to the Supplement Figures in the main text.

Reviewer 3

Thank you for the invitation to review the manuscript “Combining cross-sectional and longitudinal genomic approaches to identify determinants of cognitive and physical decline”. The authors present numerous analyses using the UK Biobank data. A lot of effort has gone into this research and the authors used a range of advanced methods, combining cross-sectional and follow-up data, using genome-wide association analysis, heritability estimation and Mendelian randomisation. The authors also compared different study design decisions and tested how these impacted their results. These considerations are important; however, the study lacks focus. Is the aim to test for differences between cross-sectional and ‘longitudinal’ genetic effects? Is this a methodological paper evaluating how best to model data from two time points? Is the aim to identify risk factors for cognitive and physical ‘decline’? Is the aim to assess the representativeness of the UK Biobank follow-up data and to test whether the results remain consistent when applying weighting? These questions are partially addressed in this manuscript, but a common thread is missing. I have included more specific comments below, which the authors may find helpful.

We appreciate the reviewer’s constructive feedback regarding the focus and organization of the manuscript. In response, we have revised the manuscript to more clearly highlight the primary aim of the study—investigating the aetiology of age-related decline within a genetically informed framework. To enhance the focus, we have moved several sensitivity analyses (e.g., evaluation of participation bias) and methodological components (e.g., simulation work) to the supplementary material. Additionally, we have expanded the theoretical background, clarified methodological definitions, and referenced relevant empirical studies to provide a more in-depth context to our analyses and findings. We hope that these revisions improve clarity and help readers better understand and interpret our findings.

R3.1 Abstract: It is difficult to view this study as one of aging ‘trajectories’. Cross-sectional data does not allow for exploration of trajectories, and neither do data from two time points.

We have now removed ‘aging trajectories’ from the text.

R3.2 I was not very familiar with the ‘physical aging’ terminology.

Similar to existing research (c.f., references below), we used the term “physical aging” to broadly describe physiological changes that occur in the human body over time, which may include declines in muscle mass, bone density, body morphology, physiological function, cellular aging (e.g., telomere shortening) or cardiovascular function.

References

- Weber, Daniela. "Differences in physical aging measured by walking speed: evidence from the English Longitudinal Study of Ageing." *BMC geriatrics* 16 (2016): 1-9.
- Shen, Jin-Bo, Yuan-Yuan Ma, and Qian Niu. "A study on the physical aging characteristics of the older people over 70 years old in China." *Frontiers in Public Health* 12 (2024): 1352894.

- Koh, S.H., Choi, S.H., Jeong, J.H., Jang, J.W., Park, K.W., Kim, E.J., Kim, H.J., Hong, J.Y., Yoon, S.J., Yoon, B. and Kang, J.H., 2020. Telomere shortening reflecting physical aging is associated with cognitive decline and dementia conversion in mild cognitive impairment due to Alzheimer's disease. *Aging (Albany NY)*, 12(5), p.4407.
- Marenberg M. Normal Physical Aging. In: *Encyclopedia of Health & Aging*. SAGE Publications, Inc. doi:10.4135/9781412956208.n163

R3.3 The abstract is lengthy and, as described above, highlights the lack of focus of the study. Only the main finding should be highlighted.

As part of the revisions to the main manuscript, we have moved several sections to the Supplement. Accordingly, we have also removed the corresponding content from the Abstract to enhance its focus and clarity.

R3.4 Introduction: Some of the language used is too colloquial (e.g., “poured into research”).

The wording has now been changed to “Substantial resources are therefore invested into research scrutinizing”.

R3.5 The FEV acronym should be defined when first used.

We have now defined the acronym in the text when first used.

R3.6 The distinction between cross-sectional and longitudinal genetic effects is not sufficiently clear and longitudinal genetic effects should be more clearly defined.

We have now provided a more detailed description of longitudinal genetic effects in the section ‘Research in context’:

[Supplement, sDiscussion]: “[...] Conceptually, genetic effects on age-related decline represent time-varying genetic effects or gene-environment interactions, where the genetic effects differ across changing environments as individuals age¹⁵. Different methodological approaches have been employed to identify age-varying genetic effects. For example, large-scale cross-sectional studies have tested for gene-by-age interactions¹⁶⁻²⁰. These studies leverage the full age spectrum available at a single point in time (e.g., ages 40 to 69 in the UK Biobank) when testing for gene-by-age interactions. However, a key limitation of cross-sectional gene-by-age designs lies in their inability to rule out potential cohort effects¹⁵, as stratifying individuals by age groups inherently stratifies groups by their year of birth. Given these and other limitations (e.g., possible non-linear age effects) associated with cross-sectional gene-by-age analyses, longitudinal GWA are considered a more robust approach for investigating time-varying genetic effects. In absence of large-scale longitudinal biobank data, early research has primarily focused on candidate genes^{21,22} or genome-wide effects on longitudinal change in smaller genotyped samples (<2000²³⁻²⁷). More recently, with the release of large samples with repeat assessments (i.e., >100,000

individuals), genome-wide studies on longitudinal changes have become more feasible, enabling the examination of genetic influences on changes in routinely collected healthcare data (e.g., BMI²⁸ or biomarker changes²⁹) and health outcomes measured through repeated research assessments.

Studying age-varying genetic effects is important for a number of reasons. First, such work allows to assess if the genetic architecture underlying lifetime levels of function is similar to that characterizing age-related change. In other words, while tests on change quantify the impact of age on genome-wide associations, cross-sectional genome-wide analyses identify marginal genetic effects that are assumed to be constant over time (c.f., study framework, **Figure 1** in the main manuscript)."

R3.6 The authors comment that prior research has documented the selective participation bias in the UK Biobank. This has also been shown for the follow-up data (e.g., Lyall et al., 2022, doi: 10.1093/braincomms/fcac119).

We thank the reviewer for pointing us to the reference, which we have included in the introduction of the manuscript:

[Manuscript, Introduction]: "Selective participation already documented at the initial recruitment stage in the UKBB^{18–20} may be exacerbated in longitudinal samples due to selective attrition and survival, where prospectively ascertained individuals represent an even healthier subset of the initially healthy²¹."

²¹ Lyall, D.M., Quinn, T., Lyall, L.M., Ward, J., Anderson, J.J., Smith, D.J., Stewart, W., Strawbridge, R.J., Bailey, M.E. and Cullen, B., 2022. Quantifying bias in psychological and physical health in the UK Biobank imaging sub-sample. *Brain communications*, 4(3), p.fcac119.

R3.7 I suggest the authors refer *differences* in physical and cognitive traits, instead of physical and cognitive aging. For the cross-sectional data by chronological age, even *decline* may not be appropriate, which would suggest changes over time.

We have intentionally used the term 'aging' to emphasize the longitudinal nature of our outcomes (i.e., changes/decline in cognitive and physical function), rather than simply referring to differences in traits. This is a common distinction made in aging research and is important for ensuring that findings on measures of aging are correctly interpreted as reflecting change rather than levels of function. As illustrated in the example below, distinguishing between ability and aging helps clarify that aging-related outcomes pertain to declines or changes over time, which can only be assessed through longitudinal data.

Reference 1: Harris, Sarah E., and Ian J. Deary. "The genetics of cognitive ability and cognitive ageing in healthy older people." *Trends in cognitive sciences* 15, no. 9 (2011): 388-394.

Cognitive ability and cognitive ageing: Consider the scores obtained by a 75-year-old individual in several mental tests; say, tests of memory, reasoning, speed of processing and so forth. This is an age at which the adult peak in such skills is typically well past. Therefore, scores on any one test will be the combination of three things: the previous peak level of the ability, how much age-associated change there has been, and occasion-specific variance (including error variance). To separate the phenotypes of level and change in cognitive abilities, longitudinal studies are required in which people have taken mental tests on more than one occasion.

Reference 2: Corley, J., Conte, F., Harris, S.E., Taylor, A.M., Redmond, P., Russ, T.C., Deary, I.J. and Cox, S.R., 2023. Predictors of longitudinal cognitive ageing from age 70 to 82 including APOE e4 status, early-life and lifestyle factors: the Lothian Birth Cohort 1936. *Molecular Psychiatry*, 28(3), pp.1256-1271.

A further challenge in understanding the predictors of cognitive ageing trajectories is the difficulty in disentangling actual cognitive change from lifelong levels of performance (which are conflated in cross-sectional data). [...] Longitudinal studies with repeated cognitive measures across an extended period in later life, paired with appropriate methodologies for modelling change, are crucial for characterising the progression of cognitive change and robustly identifying its correlates. Ideally, studies should establish the extent to which potential determinants of differences in cognitive ageing are independent of prior cognitive ability differences.

For the phenotypic age estimates obtained from between-subject (i.e., cross-sectional), we have now highlighted that this can also reflect birth cohort differences:

[Manuscript, Results]: “The largest effect of age was observed for psychomotor abilities (symbol digit substitution test), where the test performance decreased by -0.06 SD on average per additional year of age. Of note, while these results are interpreted as age effects, alternative factors (e.g., birth cohort effects⁵⁰) may also contribute to the observed between-subject differences across age groups. Further, when examining UKBB sub-samples with varying degrees of representativeness, ...”

R3.7 From an epidemiological perspective it is difficult to conceive of analyses of data from two time points as “longitudinal analyses”.

According to the definitions outlined in the epidemiological textbooks and papers cited below, our analyses using repeat measurements of the same individuals over time, even if limited to two time points, can be defined as longitudinal analyses:

Reference: Tooth, L., Ware, R., Bain, C., Purdie, D.M. and Dobson, A., 2005. Quality of reporting of observational longitudinal research. *American journal of epidemiology*, 161(3), pp.280-288.

“Longitudinal analyses were defined as those assessing change in outcome over two or more time points and that take into account the fact that the observations are likely to be

correlated.”

Reference: Reference: “Fitzmaurice, G. M., Laird, N. M., & Ware, J. H. (2012). *Applied longitudinal analysis*. John Wiley & Sons.”

“The defining feature of longitudinal studies is that measurements of the same individuals are taken repeatedly through time, thereby allowing the direct study of change over time.”

Reference: Garcia, T.P. and Marder, K., 2017. Statistical approaches to longitudinal data analysis in neurodegenerative diseases: Huntington’s disease as a model. *Current neurology and neuroscience reports*, 17(2), p.14

“Starter Methods for Longitudinal Data Analysis: When there are only two time points in the study (e.g., case 1), a straightforward approach is analyzing the change score: the differences between the measures at each time point.”

R3.8 Could the authors provide the rationale for adjusting the change score analyses for follow-up duration?

The follow-up duration was included as a covariate to derive individual slopes of change that are independent of the follow-up duration, which is important as the follow-up duration varied widely among UKB participants. A comparable approach would be to obtain annual change estimated as $\frac{Y_2 - Y_1}{t_2 - t_1}$, based on the baseline (Y0), the follow-up phenotype (Y1), the baseline date (t0) and the follow-up date (t1), but this can result in very noisy estimates when the time points are too close.

R3.9 Please state the hypotheses for the various definitions of change that were analysed.

We have now added the following with respect to the hypotheses for the three definitions of change (baseline-adjusted change, absolute change, relative change):

[Supplement, sMethods]: “One important challenge when assessing age-varying genetic effects lies in how change over time is defined. For example, interactions can be scale dependent, where an interaction may be present on one scale (e.g., absolute change) but not on another (e.g., relative change)². Additionally, adjusting for baseline differences using residual change (that is, baseline-adjusted) scores has been shown to introduce bias in observational studies when the exposure of interest is associated with baseline levels of the outcome variable³. In light of these considerations, we hypothesize the following with respect to risk of bias associated with different definitions of change:

- 1) Change adjusted for baseline values can correctly identify genetic effects on change in situations where the genetic variant is unrelated to the baseline phenotype
- 2) Change unadjusted for baseline values can correctly identify genetic effects on change even if the genetic variant is associated with baseline phenotype
- 3) Differences in findings obtained from absolute and relative change can arise due to scale dependencies”

References

²VanderWeele, T. J. *Explanation in causal inference: Methods for mediation and interaction*. (Oxford University Press, 2015).

³Glymour, M. M., Weuve, J., Berkman, L. F., Kawachi, I. & Robins, J. M. When is baseline adjustment useful in analyses of change? An example with education and cognitive change. *American journal of epidemiology* 162, 267–278 (2005).

R3.10 The analyses of sample representativeness and selective participation in the UK Biobank have in part been performed previously. This could be removed to improve focus of the study.

In response to reviewer 1, we have moved the section characterizing selective follow-up participation and the corresponding sensitivity analyses to the Supplement.

R3.11 Perhaps the term “time-invariant” is more appropriate than “cross-sectional” in distinguishing longitudinal genetic effects, i.e., those expected to differ throughout the lifespan.

‘Cross-sectional’ was mostly used as this is common terminology in genetic epidemiology when distinguishing time-varying genetic effects from those that are stable over time (c.f., references below). We have therefore included the following to clarify the definition of cross-sectional GWA in our work:

[Manuscript, Introduction]: “Large-scale biobank initiatives committed to longitudinal assessments therefore represent an unprecedented resource for the study of aging processes, enabling a move from cross-sectional (i.e., **time-invariant genetics**, such as level of grip strength) genome-wide analyses to an estimation of longitudinal (e.g., loss in grip strength from that point) genetic effects.”

References:

- Wiegrefe, Simon, et al. "Analyzing longitudinal trait trajectories using GWAS identifies genetic variants for kidney function decline." *Nature Communications* 15.1 (2024): 10061.
- Xu, Z., Shen, X., Pan, W., & Alzheimer's Disease Neuroimaging Initiative. (2014). Longitudinal analysis is more powerful than cross-sectional analysis in detecting genetic association with neuroimaging phenotypes. *PLoS one*, 9(8), e102312.
- Lee, Young, et al. "On the analysis of a repeated measure design in genome-wide association analysis." *International journal of environmental research and public health* 11.12 (2014): 12283-12303.
- Li, Xiaohui, et al. "Genome-wide linkage analysis using cross-sectional and longitudinal traits for body mass index in a subsample of the Framingham Heart Study." *BMC genetics*. Vol. 4. BioMed Central, 2003.
- Zhao, Qi, et al. "Cross-sectional and longitudinal replication analyses of genome-wide association loci of type 2 diabetes in Han Chinese." *PLoS one* 9.3 (2014): e91790.

R3.12 A limitation of describing the data presented in Figure 2 as decline is that these are cross-sectional data and differences could be due to alternative explanations (e.g., cohort effects).

We have now highlighted that the phenotypic age estimates obtained from between-subject comparisons can also reflect birth cohort differences:

[Manuscript, Results]: “The largest effect of age was observed for psychomotor abilities (symbol digit substitution test), where the test performance decreased by -0.06 SD on average per additional year of age. Of note, while these results are interpreted as age effects, alternative factors (e.g., birth cohort effects⁵⁰) may also contribute to the observed between-subject differences across age groups. Further, when examining UKBB sub-samples with varying degrees of representativeness, ...”

Reference:

⁵⁰ Glenn, N. D. (1976). Cohort analysts' futile quest: Statistical attempts to separate age, period and cohort effects. *American sociological review*, 41(5), 900-904.

R3.13 The finding that baseline-adjustment likely results in bias is of interest. How does this compare to the clinical trial literature?

This is an interesting discussion point that we believe is also of interest to the readers of this work. We have therefore elaborated more on this point in the section ‘Evaluation of two-wave models of change’ in the Supplement:

[Supplement, sDiscussion]: “Of note, inconsistencies arising from varying definitions of change have also been discussed in clinical trials employing pre-post designs⁹. In trials with balanced baseline values in the outcome variable, both absolute change and baseline-adjusted change (e.g., ANCOVA) provide unbiased estimates of the treatment effect. In such cases, ANCOVA is often preferred for its potential to increase statistical power^{10,11}. However, the assumption of baseline balance is rarely met in absence of randomization, and may still be violated under randomization due to chance imbalances¹². In those situations, baseline adjustment will provide biased results and has therefore been discouraged^{10,11,13}. The same conclusion has been drawn with respect to the use of baseline-adjusted change scores in trials on pharmacogenetic effects, where the induced bias is proportional to the baseline genetic effect (c.f., Supplementary Note 1 in Ref¹⁴). Therefore, interpreting results from baseline-adjusted models should be contingent on verifying baseline balance in clinical trials.”

References:

9. Trowman, R., Dumville, J. C., Torgerson, D. J. & Cranny, G. The impact of trial baseline imbalances should be considered in systematic reviews: A methodological case study. *Journal of clinical epidemiology* **60**, 1229–1233 (2007).
10. Oakes, J. M. & Feldman, H. A. Statistical power for nonequivalent pretest-posttest designs: The impact of change-score versus ANCOVA models. *Evaluation Review* **25**, 3–28 (2001).
11. Van Breukelen, G. J. ANCOVA versus change from baseline had more power in randomized studies and more bias in nonrandomized studies. *Journal of clinical epidemiology* **59**, 920–925 (2006).
12. Altman, D. G. Comparability of randomised groups. *Journal of the Royal Statistical Society Series D: The Statistician* **34**, 125–136 (1985).
13. Zhang, S. *et al.* Empirical comparison of four baseline covariate adjustment methods in analysis of continuous outcomes in randomized controlled trials. *Clinical epidemiology* 227–235 (2014).
14. Sadler, M. C. *et al.* Leveraging large-scale biobank EHRs to enhance pharmacogenetics of cardiometabolic disease medications. *medRxiv* (2024) doi:[10.1101/2024.04.06.24305415](https://doi.org/10.1101/2024.04.06.24305415).

R3.14 Revise “male gender” to “male sex”.

We have revised the manuscript accordingly.

R3.15 The authors use the term ‘accelerated aging’; however, it is not clearly defined in the context of this work.

We have now described the term as follows:

[Manuscript, Discussion]: “Mendelian Randomization was used to further disentangle risk factors of accelerated decline, defined as a more rapid deterioration in physical and cognitive function over time”

R3.16 The conclusion that a higher baseline function in a given trait or factors shaping that trait may not serve as a buffer against age-related decline contradicts the concept of ‘cognitive reserve’. Please discuss.

We have now discussed in more detail a number of aging theories including ‘cognitive reserve’, which is included in the section ‘research in context’:

[Supplement, sDiscussion]: “Second, genetically informed designs allow to further probe existing theoretical models of aging. For example, one key interest concerns the question as to whether ‘age is kinder to the initially more able’^{29–31}, as innate and/or early life resources may slow down age-related decline. Similar prediction are made based on theories of ‘cognitive reserve’³², ‘brain reserve’³³, ‘health capital’³⁴ or ‘differential preservation’³⁵, broadly hypothesizing that individuals with higher levels of reserve are more protected from neurobiological decay throughout the aging process. Evidence on this topic has, however, largely been mixed: While some studies suggest that health reserves may protect against age-related decline, including evidence linking educational attainment to slower cognitive decline^{36–39}, a larger body of research has failed to replicate these findings^{40–47}. Our findings are consistent with the latter body of research, suggesting that high baseline function in a given trait (e.g., cognition) or related factors (e.g., educational attainment) may not buffer against age-related decline in that trait. In other words, while cognitive or physical reserves can delay the onset of functional impairment, they are unlikely to slow the overall rate of decline.”

R3.17 Given the final paragraph highlighting the benefits of the analytical approach, what additional preventative strategies were identified using this approach?

Additional preventative targets in this context refer to risk factors that contribute to age-related decline independently of those linked to (cross-sectional) levels of function, since a targeting of those factors can further push back the age at which functional impairment

begins. An extended discussion on risk factors of decline is now included in the section 'research in context', for example:

[Supplement, Reserach in context]: "With respect to other health-related risk factors, previous observational research has mostly focused on cardiovascular risk factors^{55,56}, lipid traits⁵⁷, brain health⁵⁸, lifestyle factors (e.g., physical activity^{59,60}, alcohol use^{61,62}, smoking⁶³, diet⁶⁴), mental health (e.g., depression^{65,66}, sleep^{67,68}), early life factors (e.g., birth weight⁶⁹) or social factors (loneliness⁷⁰, lower socioeconomic status⁷¹) as possible risks involved in cognitive or physical decline. While numerous of those factors also associated with decline in our phenotypic analyses, many of those did not survive stringent covariate adjustment in MR analysis, indicating possible confounding effects. Focusing on risk factors implicated by MR, our findings converge with experimental evidence reporting possible cognition-enhancing effects of higher vegetable intake⁷² and increases in physical function following improvements in sleep and metabolic rate⁷³. Other identified risk factors reflected markers of general vulnerabilities that are not directly amenable to intervention without additional knowledge on the specific pathways involved, such as the effect of Alzheimer's liability on cognitive decline or the effect of telomere length on physical decline. Our finding that longer paternal lifespan was predictive of both reduced physical decline as well as better lifetime physical function is in line with the view that genetic and environmental factors transmitted across generations can influence the offspring's ability to maintain health during the aging process⁷⁴⁻⁷⁶."

R3.18 Notably a limitation section is missing and the discussion of findings from other studies is minimal.

We have now included a separate limitation section in the main manuscript. Further, the supplement now includes an extended limitation section to further elaborate on some of the discussion points raised in the main manuscript. Finally, a more extensive discussion of the existing evidence is now included in the section 'research in context':

[Manuscript, Discussion]: "Large-scale longitudinal biobank samples such as the UK Biobank (UKBB) have the potential to advance our understanding of the genetic and environmental contributions to aging. In this work, we exploited the prospectively ascertained UKBB sample to separate cross-sectional from longitudinal cognitive and physical function within a genome-wide framework, while evaluating the impact of design-constraints inherent to longitudinal biobank schemes. A more comprehensive discussion complementing the findings described below is available in the section 'Research in Context' in the Supplement."

[Manuscript, Discussion]: "The presented results should be interpreted in light of a number of limitations (see also Supplement for an extended discussion on the study limitations). First, age-related change was assessed using data from only two time points per individual, which reduces measurement precision compared to approaches utilizing more intensive longitudinal data. Second, despite the large sample size,

statistical power likely remains an issue. Genetic interaction effects are inherently small and harder to detect compared to marginal effects⁹⁹, which may have hindered the identification of genetic variants and causal factors of decline. Third, the predictability of age-related decline was generally low, given the number and effect sizes of the exposures identified in MR. Since our genetically informed framework only tests for lifetime risk factors, non-instrumentable time-varying environmental exposures previously implicated in age-related decline are therefore not captured in this work, examples of which include life events (e.g., loss of spouse¹⁰⁰), age-related biological changes (menopausal status¹⁰¹), toxic environmental exposures (e.g., air pollution¹⁰²) or changes in medication use¹⁰³. As such, while barriers to risk-identification may reflect insufficient power to detect small exposure effects in MR, an alternative scenario is that unmodelled time-varying environmental factors play an important role in aging processes. Finally, implementing strategies designed to increase sample representativeness (Inverse Probability Weighting, IPW), we explored the impact and direction of bias resulting from selective participation. The results implicated that selective participation influenced both phenotypic and genotypic estimates (c.f., **sDiscussion**). Therefore, boosting retention rates in future follow-up assessments will be crucial to minimize existing attrition biases. This is particularly important as statistical tools designed to probe the robustness of findings (e.g., IPW) reach a bottleneck in highly non-representative sample, where genome-wide discovery is hampered by the substantial loss in (effective) sample size when performing bias-corrected genome-wide tests.”

[Supplement, Reserach in context]: “A main goal of life course epidemiology is to elucidate the structure of the aging process and to identify factors that promote healthy aging. One central focus lies on understanding individual differences in age-related decline across different domains of functioning, as marked inter-individual variation characterizes the pace at which individuals age. In this context, genetic factors are increasingly studied, given their promise to directly (e.g., via molecular/pharmacological targets) or indirectly (e.g., via environmental targets) provide insights into aetiology and prevention.

Conceptually, genetic effects on age-related decline represent time-varying genetic effects or gene-environment interactions, where the genetic effects differ across changing environments as individuals age¹⁵. Different methodological approaches have been employed to identify age-varying genetic effects. For example, large-scale cross-sectional studies have tested for gene-by-age interactions^{16–20}. These studies leverage the full age spectrum available at a single point in time (e.g., ages 40 to 69 in the UK Biobank) when testing for gene-by-age interactions. However, a key limitation of cross-sectional gene-by-age designs lies in their inability to rule out potential cohort effects¹⁵, as stratifying individuals by age groups inherently stratifies groups by their year of birth. Given these and other limitations (e.g., possible non-linear age effects) associated with cross-sectional gene-by-age analyses, longitudinal GWA are considered a more robust approach for investigating time-varying genetic effects. In absence of large-scale longitudinal biobank data, early research has primarily focused on candidate genes^{21,22}

or genome-wide effects on longitudinal change in smaller genotyped samples (<2000²³⁻²⁷). More recently, with the release of large samples with repeat assessments (i.e., >100,000 individuals), genome-wide studies on longitudinal changes have become more feasible, enabling the examination of genetic influences on changes in routinely collected healthcare data (e.g., BMI²⁸ or biomarker changes²⁹) and health outcomes measured through repeated research assessments.

Studying age-varying genetic effects is important for a number of reasons. First, such work allows to assess if the genetic architecture underlying lifetime levels of function is similar to that characterizing age-related change. In other words, while tests on change quantify the impact of age on genome-wide associations, cross-sectional genome-wide analyses identify marginal genetic effects that are assumed to be constant over time (c.f., study framework, **Figure 1** in the main manuscript). Second, genetically informed designs allow to further probe existing theoretical models of aging. For example, one key interest concerns the question as to whether ‘age is kinder to the initially more able’³⁰⁻³², as innate and/or early life resources may slow down age-related decline. Similar prediction are made based on theories of ‘cognitive reserve’³³, ‘brain reserve’³⁴, ‘health capital’³⁵ or ‘differential preservation’³⁶, broadly hypothesizing that individuals with higher levels of reserve are more protected from neurobiological decay throughout the aging process. Evidence on this topic has, however, largely been mixed: While some studies suggest that health reserves may protect against age-related decline, including evidence linking educational attainment to slower cognitive decline³⁷⁻⁴⁰, a larger body of research has failed to replicate these findings⁴¹⁻⁴⁹. Our findings are consistent with the latter body of research, suggesting that high baseline function in a given trait (e.g., cognition) or related factors (e.g., educational attainment) may not buffer against age-related decline in that trait. In other words, while cognitive or physical reserves can delay the onset of functional impairment, they are unlikely to slow the overall rate of decline. Other lines of research have focused on common cause theories of aging, investigating whether age-related declines in physical and cognitive domains are driven by shared aetiological processes⁵⁰⁻⁵³. While small but significant associations between cognitive and physical decline have indeed been documented⁵⁰, findings remains largely inconsistent^{43,50,53-56}. In our work, we observed some indications of shared risks (e.g., shorter parental lifespan linking to both increased cognitive and physical decline) when tested in Mendelian Randomization analyses. However, most implicated risk factors demonstrated independent effects across the physical and cognitive dimensions of decline.

With respect to other health-related risk factors, previous observational research has mostly focused on cardiovascular risk factors^{57,58}, lipid traits⁵⁹, brain health⁶⁰, lifestyle factors (e.g., physical activity^{61,62}, alcohol use^{63,64}, smoking⁶⁵, diet⁶⁶), mental health (e.g., depression^{67,68}, sleep^{69,70}), early life factors (e.g., birth weight⁷¹) or social factors (loneliness⁷², lower socioeconomic status⁷³) as possible risks involved in cognitive or physical decline. While numerous of those factors also associated with decline in our phenotypic analyses, many of those did not survive stringent covariate adjustment in MR analysis, indicating possible confounding effects. Focusing on risk factors implicated by MR, our findings converge with experimental evidence reporting possible cognition-enhancing effects of higher vegetable intake⁷⁴ and increases in physical function following improvements in sleep and metabolic rate⁷⁵. Other identified risk factors

reflected markers of general vulnerabilities that are not directly amenable to intervention without additional knowledge on the specific pathways involved, such as the effect of Alzheimer's liability on cognitive decline or the effect of telomere length on physical decline. Our finding that longer paternal lifespan was predictive of both reduced physical decline as well as better lifetime physical function is in line with the view that genetic and environmental factors transmitted across generations can influence the offspring's ability to maintain health during the aging process⁷⁶⁻⁷⁸.

The presented results should be interpreted in light of a number of limitations. First, age-related change was assessed using data from only two time points per individual, which reduces measurement precision compared to approaches utilizing more intensive longitudinal data. Curating intensive longitudinal data, for example via the use of hospital record data or the implementation of additional repeat assessments in the UKBB study protocol would therefore help to refine and enhance the definitions of change used in this work. Second, despite the large sample size, statistical power likely remains an issue. Genetic interaction effects are inherently small and harder to detect compared to marginal effects⁷⁹. This limitation may also have hindered the identification of causal factors of decline in Mendelian Randomization (MR) analyses, potentially explaining why some factors were identified in phenotypic but not MR analyses. Larger studies are therefore warranted to evaluate whether these discrepancies are indicative of residual confounding in classical epidemiological research, or reflect insufficient statistical power. Third, while this study focused on commonly examined risk factors for age-related decline, further research is needed to elucidate the specific pathways through which the identified risks contribute to cognitive and physical decline over time. This is particularly important for factors that are not directly modifiable, such as sex, telomere length, parental lifespan, or genetic liability for Alzheimer's disease. Finally, the use of genetically informed designs restricts our analysis to risk factors that can be instrumented with genetic variants. Important environmental and social correlates of decline that are not instrumentable, such as air pollution⁸⁰, stressful life events^{81,82}, or housing conditions⁸³, therefore require alternative causal inference methods when put to scrutiny in observational studies."